# Learning Robust State Abstractions for Hidden-Parameter Block MDPs

**Amy Zhang**[*123]       **Shagun Sodhani**[2]       **Khimya Khetarpal**[13]       **Joelle Pineau**[123]
[1]McGill University
[2]Facebook AI Research
[3]Mila

## Abstract

Many control tasks exhibit similar dynamics that can be modeled as having common latent structure. Hidden-Parameter Markov Decision Processes (HiP-MDPs) explicitly model this structure to improve sample efficiency in multi-task settings. However, this setting makes strong assumptions on the observability of the state that limit its application in real-world scenarios with rich observation spaces. In this work, we leverage ideas of common structure from the HiP-MDP setting, and extend it to enable robust state abstractions inspired by Block MDPs. We derive instantiations of this new framework for both multi-task reinforcement learning (MTRL) and meta-reinforcement learning (Meta-RL) settings. Further, we provide transfer and generalization bounds based on task and state similarity, along with sample complexity bounds that depend on the aggregate number of samples across tasks, rather than the number of tasks, a significant improvement over prior work that use the same environment assumptions. To further demonstrate the efficacy of the proposed method, we empirically compare and show improvement over multi-task and meta-reinforcement learning baselines.

## 1 Introduction

A key open challenge in AI research that remains is how to train agents that can learn behaviors that generalize across tasks and environments. When there is common structure underlying the tasks, we have seen that multi-task reinforcement learning (MTRL), where the agent learns a set of tasks simultaneously, has definite advantages (in terms of robustness and sample efficiency) over the single-task setting, where the agent independently learns each task. There are two ways in which learning multiple tasks can accelerate learning: the agent can learn a common representation of observations, and the agent can learn a common way to behave. Prior work in MTRL has also

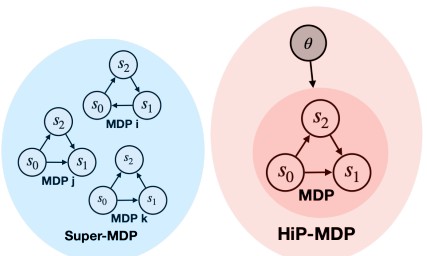

Figure 1: Visualizations of the typical MTRL setting and the HiP-MDP setting.

leveraged the idea by sharing representations across tasks (D'Eramo et al., 2020) or providing per-task sample complexity results that show improved sample efficiency from transfer (Brunskill & Li, 2013). However, explicit exploitation of the shared structure across tasks via a unified dynamics has been lacking. Prior works that make use of shared representations use a naive unification approach that posits all tasks lie in a shared domain (Figure 1, left). On the other hand, in the single-task setting, research on state abstractions has a much richer history, with several works on improved generalization through the aggregation of behaviorally similar states (Ferns et al., 2004; Li et al., 2006; Luo et al., 2019; Zhang et al., 2020b).

In this work, we propose to leverage rich state abstraction models from the single-task setting, and explore their potential for the more general multi-task setting. We frame the problem as a structured super-MDP with a shared state space and universal dynamics model conditioned on a task-specific hidden parameter (Figure 1, right). This additional structure gives us better sample efficiency, both

---

[*]Corresponding author: `amy.x.zhang@mail.mcgill.ca`

theoretically, compared to related bounds (Brunskill & Li, 2013; Tirinzoni et al., 2020) and empirically against relevant baselines (Yu et al., 2020; Rakelly et al., 2019; Chen et al., 2018; Teh et al., 2017). We learn a latent representation with smoothness properties for better few-shot generalization to other unseen tasks within this family. This allows us to derive new value loss bounds and sample complexity bounds that depend on how far away a new task is from the ones already seen.

We focus on multi-task settings where dynamics can vary across tasks, but the reward function is shared. We show that this setting can be formalized as a *hidden-parameter MDP* (HiP-MDP) (Doshi-Velez & Konidaris, 2013), where the changes in dynamics can be defined by a latent variable, unifying dynamics across tasks as a single global function. This setting assumes a global latent structure over all tasks (or MDPs). Many real-world scenarios fall under this framework, such as autonomous driving under different weather and road conditions, or even different vehicles, which change the dynamics of driving. Another example is warehouse robots, where the same tasks are performed in different conditions and warehouse layouts. The setting is also applicable to some cases of RL for medical treatment optimization, where different patient groups have different responses to treatment, yet the desired outcome is the same. With this assumed structure, we can provide concrete zero-shot generalization bounds to unseen tasks within this family. Further, we explore the setting where the state space is latent and we have access to only high-dimensional observations, and we show how to recover robust state abstractions in this setting. This is, again, a highly realistic setting in robotics when we do not always have an amenable, Lipschitz low-dimensional state space. Cameras are a convenient and inexpensive way to acquire state information, and handling pixel observations is key to approaching these problems. A *block MDP* (Du et al., 2019) provides a concrete way to formalize this observation-based setting. Leveraging this property of the block MDP framework, in combination with the assumption of a unified dynamical structure of HiP-MDPs, we introduce the *hidden-parameter block MDP* (HiP-BMDP) to handle settings with high-dimensional observations and structured, changing dynamics.

**Key contributions** of this work are a new viewpoint of the multi-task setting with same reward function as a universal MDP under the HiP-BMDP setting, which naturally leads to a gradient-based representation learning algorithm. Further, this framework allows us to compute theoretical generalization results with the incorporation of a learned state representation. Finally, empirical results show that our method outperforms other multi-task and meta-learning baselines in both fast adaptation and zero-shot transfer settings.

## 2 BACKGROUND

In this section, we introduce the base environment as well as notation and additional assumptions about the latent structure of the environments and multi-task setup considered in this work.

A finite[1], discrete-time **Markov Decision Process** (MDP) (Bellman, 1957; Puterman, 1995) is a tuple $\langle \mathcal{S}, \mathcal{A}, R, T, \gamma \rangle$, where $\mathcal{S}$ is the set of states, $\mathcal{A}$ is the set of actions, $R : \mathcal{S} \times \mathcal{A} \to \mathbb{R}$ is the reward function, $T : \mathcal{S} \times \mathcal{A} \to Dist(\mathcal{S})$ is the environment transition probability function, and $\gamma \in [0, 1)$ is the discount factor. At each time step, the learning agent perceives a state $s_t \in \mathcal{S}$, takes an action $a_t \in \mathcal{A}$ drawn from a policy $\pi : \mathcal{S} \times \mathcal{A} \to [0, 1]$, and with probability $T(s_{t+1}|s_t, a_t)$ enters next state $s_{t+1}$, receiving a numerical reward $R_{t+1}$ from the environment. The value function of policy $\pi$ is defined as: $V_\pi(s) = E_\pi[\sum_{t=0}^{\infty} \gamma^t R_{t+1}|S_0 = s]$. The optimal value function $V^*$ is the maximum value function over the class of stationary policies.

**Hidden-Parameter MDPs** (HiP-MDPs) (Doshi-Velez & Konidaris, 2013) can be defined by a tuple $\mathcal{M}$: $\langle \mathcal{S}, \mathcal{A}, \Theta, T_\theta, R, \gamma, P_\Theta \rangle$ where $\mathcal{S}$ is a finite state space, $\mathcal{A}$ a finite action space, $T_\theta$ describes the transition distribution for a specific task described by task parameter $\theta \sim P_\Theta$, $R$ is the reward function, $\gamma$ is the discount factor, and $P_\Theta$ the distribution over task parameters. This defines a family of MDPs, where each MDP is described by the parameter $\theta \sim P_\Theta$. We assume that this parameter $\theta$ is fixed for an episode and indicated by an environment id given at the start of the episode.

**Block MDPs** (Du et al., 2019) are described by a tuple $\langle \mathcal{S}, \mathcal{A}, \mathcal{X}, p, q, R \rangle$ with an unobservable state space $\mathcal{S}$, action space $\mathcal{A}$, and observable space $\mathcal{X}$. $p$ denotes the latent transition distribution $p(s'|s, a)$ for $s, s' \in \mathcal{S}, a \in \mathcal{A}$, $q$ is the (possibly stochastic) emission mapping that emits the observations $q(x|s)$ for $x \in \mathcal{X}, s \in \mathcal{S}$, and $R$ the reward function. We are interested in the setting where this

---

[1]We use this assumption only for theoretical results, but our method can be applied to continuous domains.

mapping $q$ is one-to-many. This is common in many real world problems with many tasks where the underlying states and dynamics are the same, but the observation space that the agent perceives can be quite different, e.g. navigating a house of the same layout but different decorations and furnishings.

**Assumption 1** (Block structure (Du et al., 2019)). *Each observation $x$ uniquely determines its generating state $s$. That is, the observation space $\mathcal{X}$ can be partitioned into disjoint blocks $\mathcal{X}_s$, each containing the support of the conditional distribution $q(\cdot|s)$.*

Assumption 1 gives the Markov property in $\mathcal{X}$, a key difference from partially observable MDPs (Kaelbling et al., 1998; Zhang et al., 2019), which has no guarantee of determining the generating state from the history of observations. This assumption allows us to compute reasonable bounds for our algorithm in $k$-order MDPs[2] (which describes many real world problems) and avoiding the intractability of true POMDPs, which have no guarantees on providing enough information to sufficiently predict future rewards. A relaxation of this assumption entails providing less information in the observation for predicting future reward, which will degrade performance. We show empirically that our method is still more robust to a relaxation of this assumption compared to other MTRL methods.

Bisimulation is a strict form of state abstraction, where two states are bisimilar if they are behaviorally equivalent. **Bisimulation metrics** (Ferns et al., 2011) define a distance between states as follows:

**Definition 1** (Bisimulation Metric (Theorem 2.6 in Ferns et al. (2011))). *Let $(\mathcal{S}, \mathcal{A}, P, r)$ be a finite MDP and $\mathfrak{met}$ the space of bounded pseudometrics on $\mathcal{S}$ equipped with the metric induced by the uniform norm. Define $F : \mathfrak{met} \mapsto \mathfrak{met}$ by*

$$F(d)(s, s') = \max_{a \in \mathcal{A}}(|r_s^a - r_{s'}^a| + \gamma W(d)(P_s^a, P_{s'}^a)),$$

*where $W(d)$ is the Wasserstein distance between transition probability distributions. Then $F$ has a unique fixed point $\tilde{d}$ which is the bisimulation metric.*

A nice property of this metric $\tilde{d}$ is that difference in optimal value between two states is bounded by their distance as defined by this metric.

**Theorem 1** ($V^*$ is Lipschitz with respect to $\tilde{d}$ (Ferns et al., 2004)). *Let $V^*$ be the optimal value function for a given discount factor $\gamma$. Then $V^*$ is Lipschitz continuous with respect to $\tilde{d}$ with Lipschitz constant $\frac{1}{1-\gamma}$,*

$$|V^*(s) - V^*(s')| \leq \frac{1}{1 - \gamma}\tilde{d}(s, s').$$

Therefore, we see that bisimulation metrics give us a Lipschitz value function with respect to $\tilde{d}$.

For downstream evaluation of the representations we learn, we use **Soft Actor Critic** (SAC) (Haarnoja et al., 2018), an off-policy actor-critic method that uses the maximum entropy framework for soft policy iteration. At each iteration, SAC performs soft policy evaluation and improvement steps. The policy evaluation step fits a parametric soft Q-function $Q(s_t, a_t)$ using transitions sampled from the replay buffer $\mathcal{D}$ by minimizing the soft Bellman residual,

$$J(Q) = \mathbb{E}_{(s_t, s_t, r_t, s_{t+1}) \sim \mathcal{D}}\left[\left(Q(s_t, a_t) - r_t - \gamma \bar{V}(x_{t+1})\right)^2\right].$$

The target value function $\bar{V}$ is approximated via a Monte-Carlo estimate of the following expectation,

$$\bar{V}(x_{t+1}) = \mathbb{E}_{a_{t+1} \sim \pi}\left[\bar{Q}(x_{t+1}, a_{t+1}) - \alpha \log \pi(a_{t+1}|s_{t+1})\right],$$

where $\bar{Q}$ is the target soft Q-function parameterized by a weight vector obtained from an exponentially moving average of the Q-function weights to stabilize training. The policy improvement step then attempts to project a parametric policy $\pi(a_t|s_t)$ by minimizing KL divergence between the policy and a Boltzmann distribution induced by the Q-function, producing the following objective,

$$J(\pi) = \mathbb{E}_{s_t \sim \mathcal{D}}\left[\mathbb{E}_{a_t \sim \pi}[\alpha \log(\pi(a_t|s_t)) - Q(s_t, a_t)]\right].$$

---

[2]Any $k$-order MDP can be made Markov by stacking the previous $k$ observations and actions together.

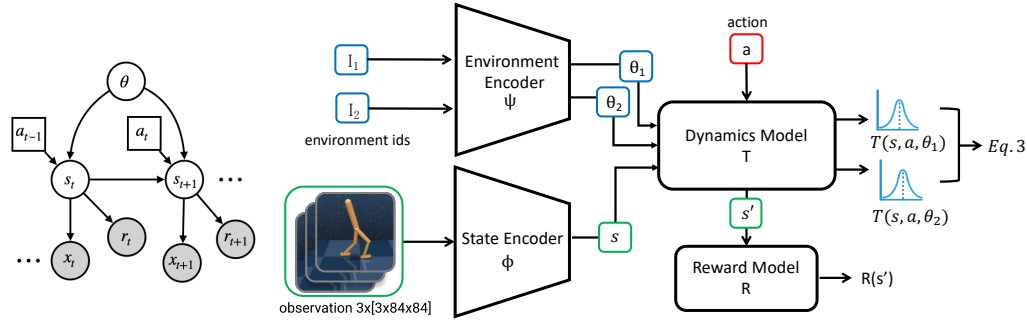

Figure 2: Graphical model of HiP-BMDP setting (left). Flow diagram of learning a HiP-BMDP (right). Two environment ids are selected by permuting a randomly sampled batch of data from the replay buffer, and the loss objective requires computing the Wasserstein distance of the predicted next-step distribution for those states.

## 3 THE HIP-BMDP SETTING

The HiP-MDP setting (as defined in Section 2) assumes full observability of the state space. However, in most real-world scenarios, we only have access to high-dimensional, noisy observations, which often contain irrelevant information to the reward. We combine the Block MDP and HiP-MDP settings to introduce the **Hidden-Parameter Block MDP** setting (HiP-BMDP), where states are latent, and transition distributions change depending on the task parameters $\theta$. This adds an additional dimension of complexity to our problem – we first want to learn an amenable state space $\mathcal{S}$, and a universal dynamics model in that representation[3]. In this section, we formally define the HiP-BMDP family in Section 3.1, propose an algorithm for learning HiP-BMDPs in Section 3.2, and finally provide theoretical analysis for the setting in Section 3.3.

### 3.1 THE MODEL

A HiP-BMDP family can be described by tuple $\langle \mathcal{S}, \mathcal{A}, \Theta, T_\theta, R, \gamma, P_\Theta, \mathcal{X}, q \rangle$, with a graphical model of the framework found in Figure 2. We are given a label $k \in \{1, ..., N\}$ for each of $N$ environments. We plan to learn a candidate $\Theta$ that unifies the transition dynamics across all environments, effectively finding $T(\cdot, \cdot, \theta)$. For two environment settings $\theta_i, \theta_j \in \Theta$, we define a distance metric:

$$d(\theta_i, \theta_j) := \max_{s,a \in \{S,A\}} \left[ W\big(T_{\theta_i}(s,a), T_{\theta_j}(s,a)\big) \right]. \tag{1}$$

The Wasserstein-1 metric can be written as $W_d(P,Q) = \sup_{f \in \mathcal{F}_d} \big\| \mathbb{E}_{x \sim P} f(x) - \mathbb{E}_{y \sim Q} f(y) \big\|_1$, where $\mathcal{F}_d$ is the set of 1-Lipschitz functions under metric $d$ (Müller, 1997). We omit $d$ but use $d(x,y) = \|x - y\|_1$ in our setting. This ties distance between $\theta$ to the maximum difference in the next state distribution of all state-action pairs in the MDP.

Given a HiP-BMDP family $\mathcal{M}_\Theta$, we assume a multi-task setting where environments with specific $\theta \in \Theta$ are sampled from this family. We do not have access to $\theta$, and instead get environment labels $I_1, I_2, ..., I_N$. The goal is to learn a latent space for the hyperparameters $\theta$[4]. We want $\theta$ to be smooth with respect to changes in dynamics from environment to environment, which we can set explicitly through the following objective:

$$\|\psi(I_1) - \psi(I_2)\|_1 = \max_{\substack{s \in \mathcal{S} \\ a \in \mathcal{A}}} \left[ W_2\big(p(s_{t+1}|s_t, a_t, \psi(I_1)), p(s_{t+1}|s_t, a_t, \psi(I_2))\big) \right], \tag{2}$$

given environment labels $I_1, I_2$ and $\psi : \mathbb{Z}^+ \mapsto \mathbb{R}^d$, the encoder that maps from environment label, the set of positive integers, to $\theta$.

### 3.2 LEARNING HIP-BMDPS

The premise of our work is that the HiP-BMDP formulation will improve sample efficiency and generalization performance on downstream tasks. We examine two settings, multi-task reinforcement

---

[3]We overload notation here since the true state space is latent.

[4]We again overload notation here to refer to the learned hyperparameters as $\theta$, as the true ones are latent.

learning (MTRL) and meta-reinforcement learning (meta-RL). In both settings, we have access to $N$ training environments and a held-out set of $M$ evaluation environments, both drawn from a defined family. In the MTRL setting, we evaluate model performance across all $N$ training environments and ability to adapt to new environments. Adaptation performance is evaluated in both the few-shot regime, where we collect a small number of samples from the evaluation environments to learn each hidden parameter $\theta$, and the zero-shot regime, where we average $\theta$ over all training tasks. We evaluate against ablations and other MTRL methods. In the meta-RL setting, the goal for the agent is to leverage knowledge acquired from the previous tasks to adapt quickly to a new task. We evaluate performance in terms of how quickly the agent can achieve a minimum threshold score in the unseen evaluation environments (by learning the correct $\theta$ for each new environment).

Learning a HiP-BMDP approximation of a family of MDPs requires the following components: **i)** an encoder that maps observations from state space to a learned, latent representation, $\phi : \mathcal{X} \mapsto \mathcal{Z}$, **ii)** an environment encoder $\psi$ that maps an environment identifier to a hidden parameter $\theta$, **iii)** a universal dynamics model $T$ conditioned on task parameter $\theta$. Figure 2 shows how the components interact during training. In practice, computing the maximum Wasserstein distance over the entire state-action space is computationally infeasible. Therefore, we relax this requirement by taking the expectation over Wasserstein distance with respect to the marginal state distribution of the behavior policy. We train a probabilistic universal dynamics model $T$ to output the desired next state distributions as Gaussians[5], for which the 2-Wasserstein distance has a closed form:

$$W_2(\mathcal{N}(m_1, \Sigma_1), \mathcal{N}(m_2, \Sigma_2))^2 = ||m_1 - m_2||_2^2 + ||\Sigma_1^{1/2} - \Sigma_2^{1/2}||_{\mathcal{F}}^2,$$

where $|| \cdot ||_{\mathcal{F}}$ is the Frobenius norm.

Given that we do not have access to the true universal dynamics function across all environments, it must be learned. The objective in Equation (2) is accompanied by an additional objective to learn $T$, giving a final loss function:

$$\mathcal{L}(\psi, T) = MSE \underbrace{\left( \Big| \Big| \psi(I_1) - \psi(I_2) \Big| \Big|_2, W_2 \big( T(s_t^{I_1}, \pi(s_t^{I_1}), \psi(I_1)), T(s_t^{I_2}, \pi(s_t^{I_2}), \psi(I_2)) \big) \right)}_{\Theta \text{ learning error}}$$

$$+ MSE \underbrace{\left( T(s_t^{I_1}, a_t^{I_1}, \psi(I_1)), s_{t+1}^{I_1} \right) + MSE \left( T(s_t^{I_2}, a_t^{I_2}, \psi(I_2)), s_{t+1}^{I_2} \right)}_{\text{Model learning error}}. \tag{3}$$

where red indicates gradients are stopped. Transitions $\{s_t^{I_1}, a_t^{I_1}, s_{t+1}^{I_1}, I_1\}$ and $\{s_t^{I_2}, a_t^{I_2}, s_{t+1}^{I_2}, I_2\}$ from two different environments ($I_1 \neq I_2$) are sampled randomly from a replay buffer. In practice, we scale the $\Theta$ learning error, our task bisimulation metric loss, using a scalar value denoted as $\alpha_\psi$.

### 3.3 THEORETICAL ANALYSIS

In this section, we provide value bounds and sample complexity analysis of the HiP-BMDP approach. We have additional new theoretical analysis of the simpler HiP-MDP setting in Appendix B. We first define three additional error terms associated with learning a $\epsilon_R, \epsilon_T, \epsilon_\theta$-bisimulation abstraction,

$$\epsilon_R := \sup_{\substack{a \in \mathcal{A}, \\ x_1, x_2 \in \mathcal{X}, \phi(x_1) = \phi(x_2)}} \big| R(x_1, a) - R(x_2, a) \big|,$$

$$\epsilon_T := \sup_{\substack{a \in \mathcal{A}, \\ x_1, x_2 \in \mathcal{X}, \phi(x_1) = \phi(x_2)}} \big\| \Phi T(x_1, a) - \Phi T(x_2, a) \big\|_1,$$

$$\epsilon_\theta := \|\hat{\theta} - \theta\|_1.$$

$\Phi T$ denotes the *lifted* version of $T$, where we take the next-step transition distribution from observation space $\mathcal{X}$ and lift it to latent space $\mathcal{S}$. We can think of $\epsilon_R, \epsilon_T$ as describing a new MDP which is close – but not necessarily the same, if $\epsilon_R, \epsilon_T > 0$ – to the original Block MDP. These two error terms can be computed empirically over all training environments and are therefore not task-specific.

---

[5]This is not a restrictive assumption to make, as any distribution can be mapped to a Gaussian with an encoder of sufficient capacity.

$\epsilon_\theta$, on the other hand, is measured as a per-task error. Similar methods are used in Jiang et al. (2015) to bound the loss of a single abstraction, which we extend to the HiP-BMDP setting with a family of tasks.

**Value Bounds.** We first evaluate how the error in $\theta$ prediction and the learned bisimulation representation affect the optimal $Q^*_{\bar{\mathcal{M}}_{\hat{\theta}}}$ of the learned MDP, by first bounding its distance from the optimal $Q^*$ of the true MDP for a single-task.

**Theorem 2** (Q error). *Given an MDP $\bar{\mathcal{M}}_{\hat{\theta}}$ built on a $(\epsilon_R, \epsilon_T, \epsilon_\theta)$-approximate bisimulation abstraction of an instance of a HiP-BMDP $\mathcal{M}_\theta$, we denote the evaluation of the optimal Q function of $\bar{\mathcal{M}}_{\hat{\theta}}$ on $\mathcal{M}$ as $[Q^*_{\bar{\mathcal{M}}_{\hat{\theta}}}]_{\mathcal{M}_\theta}$. The value difference with respect to the optimal $Q^*_{\mathcal{M}}$ is upper bounded by*

$$\left\| Q^*_{\mathcal{M}_\theta} - [Q^*_{\bar{\mathcal{M}}_{\hat{\theta}}}]_{\mathcal{M}_\theta} \right\|_\infty \leq \epsilon_R + \gamma(\epsilon_T + \epsilon_\theta)\frac{R_{max}}{2(1-\gamma)}.$$

Proof in Appendix C. As in the HiP-MDP setting, we can measure the transferability of a specific policy $\pi$ learned on one task to another, now taking into account error from the learned representation.

**Theorem 3** (Transfer bound). *Given two MDPs $\mathcal{M}_{\theta_i}$ and $\mathcal{M}_{\theta_j}$, we can bound the difference in $Q^\pi$ between the two MDPs for a given policy $\pi$ learned under an $\epsilon_R, \epsilon_T, \epsilon_{\theta_i}$-approximate abstraction of $\mathcal{M}_{\theta_i}$ and applied to*

$$\left\| Q^*_{\mathcal{M}_{\theta_j}} - [Q^*_{\bar{\mathcal{M}}_{\hat{\theta}_i}}]_{\mathcal{M}_{\theta_j}} \right\|_\infty \leq \epsilon_R + \gamma\left(\epsilon_T + \epsilon_{\theta_i} + \|\theta_i - \theta_j\|_1\right)\frac{R_{max}}{2(1-\gamma)}.$$

This result clearly follows directly from Theorem 2. Given a policy learned for task $i$, Theorem 3 gives a bound on how far from optimal that policy is when applied to task $j$. Intuitively, the more similar in behavior tasks $i$ and $j$ are, as denoted by $\|\theta_i - \theta_j\|_1$, the better $\pi$ performs on task $j$.

**Finite Sample Analysis.** In MDPs (or families of MDPs) with large state spaces, it can be unrealistic to assume that all states are visited at least once, in the finite sample regime. Abstractions are useful in this regime for their generalization capabilities. We can instead perform a counting analysis based on the number of samples of any abstract state-action pair.

We compute a loss bound with abstraction $\phi$ which depends on the size of the replay buffer $D$, collected over all tasks. Specifically, we define the minimal number of visits to an abstract state-action pair, $n_\phi(D) = \min_{x \in \phi(\mathcal{S}), a \in \mathcal{A}} |D_{x,a}|$. This sample complexity bound relies on a Hoeffding-style inequality, and therefore requires that the samples in $D$ be independent, which is usually not the case when trajectories are sampled.

**Theorem 4** (Sample Complexity). *For any $\phi$ which defines an $(\epsilon_R, \epsilon_T, \epsilon_\theta)$-approximate bisimulation abstraction on a HiP-BMDP family $\mathcal{M}_\Theta$, we define the empirical measurement of $Q^*_{\bar{\mathcal{M}}_{\hat{\theta}}}$ over $D$ to be $Q^*_{\bar{\mathcal{M}}_{\hat{\theta}}^D}$. Then, with probability $\geq 1 - \delta$,*

$$\left\| Q^*_{\mathcal{M}_\theta} - [Q^*_{\bar{\mathcal{M}}_{\hat{\theta}}^D}]_{\mathcal{M}_\theta} \right\|_\infty \leq \epsilon_R + \gamma(\epsilon_T + \epsilon_\theta)\frac{R_{max}}{2(1-\gamma)} + \frac{R_{max}}{(1-\gamma)^2}\sqrt{\frac{1}{2n_\phi(D)}\log\frac{2|\phi(\mathcal{X})||\mathcal{A}|}{\delta}}. \quad (4)$$

This performance bound applies to all tasks in the family and has two terms that are affected by using a state abstraction: the number of samples $n_\phi(D)$, and the size of the state space $|\phi(\mathcal{X})|$. We know that $|\phi(\mathcal{X})| \leq |\mathcal{X}|$ as behaviorally equivalent states are grouped together under bisimulation, and $n_\phi(D)$ is the minimal number of visits to any abstract state-action pair, in aggregate over all training environments. This is an improvement over the sample complexity of applying single-task learning without transfer over all tasks, and the method proposed in Brunskill & Li (2013), which both would rely on the number of tasks or number of MDPs seen.

## 4 EXPERIMENTS & RESULTS

We use environments from Deepmind Control Suite (DMC) (Tassa et al., 2018) to evaluate our method for learning HiP-BMDPs for both multi-task RL and meta-reinforcement learning settings.

We consider two setups for evaluation: **i)** an *interpolation* setup and **ii)** an *extrapolation* setup where the changes in the dynamics function are interpolations and extrapolations between the changes in the dynamics function of the training environment respectively. This dual-evaluation setup provides a more nuanced understanding of how well the learned model transfers across the environments. Implementation details can be found in Appendix D and sample videos of policies at `https://sites.google.com/view/hip-bmdp`.

**Environments.** We create a family of MDPs using the existing environment-task pairs from DMC and change one environment parameter to sample different MDPs. We denote this parameter as the *perturbation*-parameter. We consider the following HiP-BMDPs: 1. `Cartpole-Swingup-V0`: the mass of the pole varies, 2. `Cheetah-Run-V0`: the size of the torso varies, 3.



Figure 3: Variation in `Cheetah-Run-V0` tasks.

`Walker-Run-V0`: the friction coefficient between the ground and the walker's legs varies, 4. `Walker-Run-V1`: the size of left-foot of the walker varies, and 5. `Finger-Spin-V0`: the size of the finger varies. We show an example of the different pixel observations for `Cheetah-Run-V0` in Figure 3. Additional environment details are in Appendix D.

We sample 8 MDPs from each MDP family by sampling different values for the *perturbation*-parameter. The MDPs are arranged in order of increasing values of the *perturbation*-parameter such that we can induce an order over the family of MDPs. We denote the ordered MDPs as $A - H$. MDPs $\{B, C, F, G\}$ are training environments and $\{D, E\}$ are used for evaluating the model in the interpolation setup (i.e. the value of the *perturbation*-parameter can be obtained by interpolation). MDPs $\{A, H\}$ are for evaluating the model in the extrapolation setup (i.e. the value of the *perturbation*-parameter can be obtained by extrapolation). We evaluate the learning agents by computing average reward (over 10 episodes) achieved by the policy after training for a fixed number of steps. All experiments are run for 10 seeds, with mean and standard error reported in the plots.

**Multi-Task Setting.** We first consider a multi-task setup where the agent is trained on four related, but different environments with pixel observations. We compare our method, HiP-BMDP, with the following baselines and ablations: **i)** *DeepMDP* (Gelada et al., 2019) where we aggregate data across all training environments, **ii)** *HiP-BMDP-nobisim*, HiP-BMDP without the task bisimulation metric loss on task embeddings, iii) *Distral*, an ensemble of policies trained using the Distral algorithm (Teh et al., 2017) with *SAC-AE* (Yarats et al., 2019) as the underlying policy, iv) PCGrad (Yu et al., 2020), and v) GradNorm (Chen et al., 2018). For all models, the agent sequentially performs one update per environment. For fair comparison, we ensure that baselines have at least as many parameters as HiP-BMDP. *Distral* has more parameters as it trains one policy per environment. Additional implementation details about baselines are in Appendix D.1.

In Figures 4, and 10 (in Appendix), we observe that for all the models, performance deteriorates when evaluated on interpolation/extrapolation environments. We only report extrapolation results in the main paper because of space constraints, as they were very similar to the interpolation performance. The gap between the HiP-BMDP model and other baselines also widens, showing that the proposed approach is relatively more robust to changes in environment dynamics.

At training time (Figure 9 in Appendix), we observe that HiP-BMDP consistently outperforms other baselines on all the environments. The success of our proposed method can not be attributed to task embeddings alone as HiP-BMDP-nobisim also uses task embeddings. Moreover, only incorporating the task-embeddings is not guaranteed to improve performance in all the environments (as can be seen in the case of `Cheetah-Run-V0`). We also note that the multi-task learning baselines like Distral, PCGrad, and GradNorm sometimes lag behind even the DeepMDP baseline, perhaps because they do not leverage a shared global dynamics model.

**Meta-RL Setting.** We consider the Meta-RL setup for evaluating the few-shot generalization capabilities of our proposed approach on proprioceptive state, as meta-RL techniques are too time-intensive to train on pixel observations directly. Specifically, we use PEARL (Rakelly et al., 2019), an off-policy meta-learning algorithm that uses probabilistic context variables, and is shown to outperform common meta-RL baselines like MAML-TRPO (Finn et al., 2017) and ProMP (Rothfuss et al., 2019) on proprioceptive state. We incorporate our proposed approach in PEARL by training the *inference network* $q_\phi(\mathbf{z}|\mathbf{c})$ with our additional HiP-BMDP loss. The algorithm pseudocode can be found in Appendix D. In Figure 5 we see that the proposed approach (blue) converges faster to a

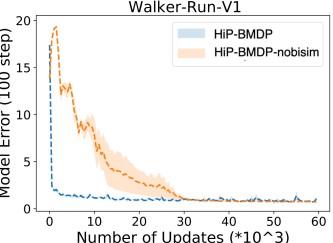

Figure 4: Multi-Task Setting. Zero-shot generalization performance on the extrapolation tasks. We see that our method, HiP-BMDP, performs best against all baselines across all environments. Note that the environment steps (on the x-axis) denote the environment steps for each task. Since we are training over four environments, the actual number of steps is approx. 3.2 million.

threshold reward (green) than the baseline for `Cartpole-Swingup-V0` and `Walker-Walk-V1`. We provide additional results in Appendix E.

**Evaluating the Universal Transition Model.** We investigate how well the transition model performs in an unseen environment by only adapting the task parameter $\theta$. We instantiate a new MDP, sampled from the family of MDPs, and use a behavior policy to collect transitions. These transitions are used to update only the $\theta$ parameter, and the transition model is evaluated by unrolling the transition model for $k$-steps. We report the average, per-step model error in latent space, averaged over 10 environments. While we expect both the proposed setup and baseline setups to adapt to the new environment, we expect the proposed setup to adapt faster because of the exploitation of underlying structure. In Figure 6, we indeed observe that the proposed *HiP-BMDP* model adapts much faster than the ablation *HiP-BMDP-nobisim*.

Figure 6: Average per-step model error (in latent space) after unrolling the transition model for 100 steps.

**Relaxing the Block MDP Assumption.** We incorporate *sticky observations* into the environment to determine how HiP-BMDP behaves when the Block MDP assumption is relaxed. For some probability $p$ (set to 0.1 in practice), the current observation is dropped, and the agent sees the previous observation again. In Figure 7, we see that even in this setting the proposed HiP-BMDP model outperforms the other baseline models.

## 5 RELATED WORK

Multi-task learning has been extensively studied in RL with assumptions around common properties of different tasks, e.g., reward and transition dynamics. A lot of work has focused on considering tasks as MDPs and learning optimal policies for each task while maximizing shared knowledge. However, in most real-world scenarios, the parameters governing the dynamics are not observed. Moreover, it is not explicitly clear how changes in dynamics across tasks are controlled. The HiP-BMDP setting provides a principled way to change dynamics across tasks via a latent variable.

Much existing work in the **multi-task reinforcement learning** (MTRL) setting focuses on learning shared representations (Ammar et al., 2014; Parisotto et al., 2016; Calandriello et al., 2014; Maurer et al., 2016; Landolfi et al., 2019). D'Eramo et al. (2020) extend approximate value iteration bounds in the single-task setting to the multi-task by computing the average loss across tasks and Brunskill & Li (2013) offer sample complexity results, which still depend on the number of tasks, unlike ours. Sun et al. (2020); Tirinzoni et al. (2020) also obtain PAC bounds on sample complexity for the MTRL setting, but Sun et al. (2020) relies on a constructed state-action abstraction that assumes

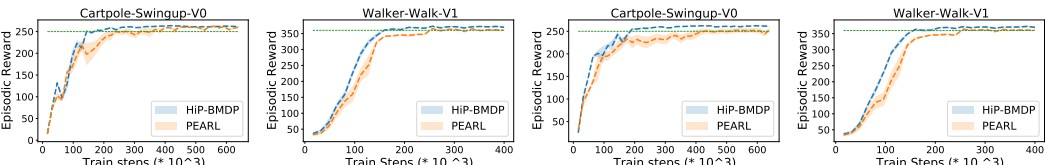

Figure 5: Few-shot generalization performance on the interpolation (2 left) and extrapolation (2 right) tasks. Green line shows a threshold reward. 100 steps are used for adaptation to the evaluation environments.

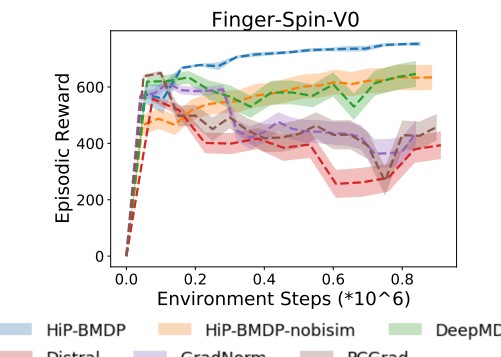 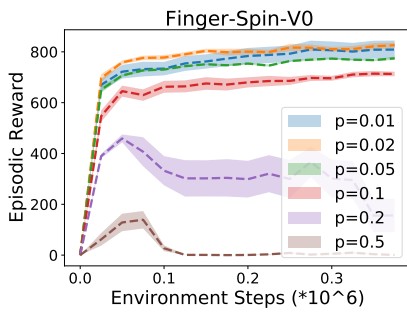

Figure 7: Zero-shot generalization performance on the evaluation tasks in the MTRL setting with partial observability. HiP-BMDP (ours) consistently outperforms other baselines (left). We also show decreasing performance by HiP-BMDP as $p$ increases (right). 10 seeds, 1 stderr shaded.

a discrete and tractably small state space. Tirinzoni et al. (2020) assumes access to a generative model for any state-action pair and scales with the minimum of number of tasks or state space. In the rich observation setting, this minimum will almost always be the number of tasks. Similar to our work, Perez et al. (2020) also treats the multi-task setting as a HiP-MDP by explicitly designing latent variable models to model the latent parameters, but require knowledge of the structure upfront, whereas our approach does not make any such assumptions.

**Meta-learning**, or learning to learn, is also a related framework with a different approach. We focus here on context-based approaches, which are more similar to the shared representation approaches of MTRL and our own method. Rakelly et al. (2019) model and learn latent contexts upon which a universal policy is conditioned. However, no explicit assumption of a universal structure is leveraged. Amit & Meir (2018); Yin et al. (2020) give a PAC-Bayes bound for meta-learning generalization that relies on the number of tasks $n$. Our setting is quite different from the typical assumptions of the meta-learning framework, which stresses that the tasks must be mutually exclusive to ensure a single model cannot solve all tasks. Instead, we assume a shared latent structure underlying all tasks, and seek to exploit that structure for generalization. We find that under this setting, our method indeed outperforms policies initialized through meta-learning.

The ability to extract meaningful information through **state abstractions** provides a means to generalize across tasks with a common structure. Abel et al. (2018) learn transitive and PAC state abstractions for a distribution over tasks, but they concentrate on finite, tabular MDPs. One approach to form such abstractions is via **bisimulation metrics** (Givan et al., 2003; Ferns et al., 2004) which formalize a concrete way to group behaviorally equivalent states. Prior work also leverages bisimulation for transfer (Castro & Precup, 2010), but on the policy level. Our work instead focuses on learning a latent state representation and established theoretical results for the MTRL setting. Recent work (Gelada et al., 2019) also learns a latent dynamics model and demonstrates connections to bisimulation metrics, but does not address multi-task learning.

## 6 DISCUSSION

In this work, we advocate for a new framework, HiP-BMDP, to address the multi-task reinforcement learning setting. Like previous methods, HiP-BMDP assumes a shared state and action space across tasks, but additionally assumes latent structure in the dynamics. We exploit this structure through learning a universal dynamics model with latent parameter $\theta$, which captures the behavioral similarity across tasks. We provide error and value bounds for the HiP-MDP (in appendix) and HiP-BMDP settings, showing improvements in sample complexity over prior work by producing a bound that depends on the number of samples in aggregate over tasks, rather than number of tasks seen at training time. Our work relies on an assumption that we have access to an environment id, or knowledge of when we have switched environments. This assumption could be relaxed by incorporating an environment identification procedure at training time to cluster incoming data into separate environments. Further, our bounds rely $L^\infty$ norms for measuring error and the value and transfer bounds. In future work we will investigate tightening these bounds with $L^p$ norms.

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

## A BISIMULATION BOUNDS

We first look at the Block MDP case only (Zhang et al., 2020a), which can be thought of as the single-task setting in a HiP-BMDP. We can compute approximate error bounds in this setting by denoting $\phi$ an $(\epsilon_R, \epsilon_T)$-approximate bisimulation abstraction, where

$$\epsilon_R := \sup_{\substack{a \in \mathcal{A}, \\ x_1, x_2 \in \mathcal{X}, \phi(x_1) = \phi(x_2)}} \big| R(x_1, a) - R(x_2, a) \big|,$$

$$\epsilon_T := \sup_{\substack{a \in \mathcal{A}, \\ x_1, x_2 \in \mathcal{X}, \phi(x_1) = \phi(x_2)}} \big\| \Phi T(x_1, a) - \Phi T(x_2, a) \big\|_1.$$

$\Phi T$ denotes the *lifted* version of $T$, where we take the next-step transition distribution from observation space $\mathcal{X}$ and lift it to latent space $\mathcal{S}$.

**Theorem 5.** *Given an MDP $\bar{\mathcal{M}}$ built on a $(\epsilon_R, \epsilon_T)$-approximate bisimulation abstraction of Block MDP $\mathcal{M}$, we denote the evaluation of the optimal Q function of $\bar{\mathcal{M}}$ on $\mathcal{M}$ as $[Q^*_{\bar{\mathcal{M}}}]_{\mathcal{M}}$. The value difference with respect to the optimal $Q^*_{\mathcal{M}}$ is upper bounded by*

$$\big\| Q^*_{\mathcal{M}} - [Q^*_{\bar{\mathcal{M}}}]_{\mathcal{M}} \big\|_\infty \leq \epsilon_R + \gamma \epsilon_T \frac{R_{max}}{2(1 - \gamma)}.$$

*Proof.* From Theorem 2 in Jiang (2018). $\qquad\square$

## B THEORETICAL RESULTS FOR THE HiP-MDP SETTING

We explore the HiP-MDP setting, where a low-dimensional state space is given, to highlight the results that can be obtained just from assuming this hierarchical structure of the dynamics.

### B.1 VALUE BOUNDS

Given a family of environments $\mathcal{M}_\Theta$, we bound the difference in expected value between two sampled MDPs, $\mathcal{M}_{\theta_i}, \mathcal{M}_{\theta_j} \in \mathcal{M}_\Theta$ using $d(\theta_i, \theta_j)$. Additionally, we make the assumption that we have a behavior policy $\pi$ that is near both optimal policies $\pi^*_{\theta_i}, \pi^*_{\theta_j}$. We use KL divergence to define this neighborhood for $\pi^*_{\theta_i}$,

$$d^{\text{KL}}(\pi, \pi^*_{\theta_i}) = \mathbb{E}_{s \sim \rho^\pi} \big[ KL(\pi(\cdot|s), \pi^*_{\theta_i}(\cdot|s))^{1/2} \big]. \tag{5}$$

We start with a bound for a specific policy $\pi$. One way to measure the difference between two tasks $\mathcal{M}_{\theta_i}, \mathcal{M}_{\theta_j}$ is to measure the difference in value when that policy is applied in both settings. We show the relationship between the learned $\theta$ and this difference in value. The following results are similar to error bounds in approximate value iteration (Munos, 2005; Bertsekas & Tsitsiklis, 1996), but instead of tracking model error, we apply these methods to compare tasks with differences in dynamics.

**Theorem 6.** *Given policy $\pi$, the difference in expected value between two MDPs drawn from the family of MDPs $\mathcal{M}_{\theta_i}, \mathcal{M}_{\theta_j} \in \mathcal{M}_\Theta$ is bounded by*

$$|V^\pi_{\theta_i} - V^\pi_{\theta_j}| \leq \frac{\gamma}{1 - \gamma} \|\theta_i - \theta_j\|_1. \tag{6}$$

*Proof.* We use a telescoping sum to prove this bound, which is similar to Luo et al. (2019). First, we let $Z_k$ denote the discounted sum of rewards if the first $k$ steps are in $\mathcal{M}_{\theta_i}$, and all steps $t > k$ are in $\mathcal{M}_{\theta_j}$,

$$Z_k := \mathbb{E}_{\substack{\forall t \geq 0, a_t \sim \pi(s_t) \\ \forall j > t \geq 0, s_{t+1} \sim T_{\theta_i}(s_t, a_t) \\ \forall t \geq j, s_{t+1} \sim T_{\theta_j}(s_t, a_t)}} \bigg[ \sum_{t=0}^\infty \gamma^t R(s_t, a_t) \bigg].$$

By definition, we have $Z_\infty = V^\pi_{\theta_i}$ and $Z_0 = V^\pi_{\theta_j}$. Now, the value function difference can be written as a telescoping sum,

$$V^\pi_{\theta_i} - V^\pi_{\theta_j} = \sum_{k=0}^\infty (Z_{k+1} - Z_k). \tag{7}$$

Each term can be simplified to

$$Z_{k+1} - Z_k = \gamma^{k+1} \mathbb{E}_{s_k, a_k \sim \pi, T_{\theta_i}} \left[ \mathbb{E}_{\substack{s_{k+1} \sim T_{\theta_j}(\cdot|s_k, a_k), \\ s'_{k+1} \sim T_{\theta_i}(\cdot|s_k, a_k)}} \left[ V_{\theta_j}^\pi(s_{k+1}) - V_{\theta_j}^\pi(s'_{k+1}) \right] \right].$$

Plugging this back into Equation (7),

$$V_{\theta_i}^\pi - V_{\theta_j}^\pi = \frac{\gamma}{1-\gamma} \mathbb{E}_{\substack{s \sim \rho_{\theta_i}^\pi, \\ a \sim \pi(s)}} \left[ \mathbb{E}_{s' \sim T_{\theta_i}(\cdot|s,a)} V_{\theta_j}^\pi(s') - \mathbb{E}_{s' \sim T_{\theta_j}(\cdot|s,a)} V_{\theta_j}^\pi(s') \right].$$

This expected value difference is bounded by the Wasserstein distance between $T_{\theta_i}, T_{\theta_j}$,

$$|V_{\theta_i}^\pi - V_{\theta_j}^\pi| \leq \frac{\gamma}{1-\gamma} W(T_{\theta_i}, T_{\theta_j})$$

$$= \frac{\gamma}{1-\gamma} \|\theta_i - \theta_j\|_1 \quad \text{using Equation (1).}$$

$\square$

Another comparison to make is how different the optimal policies in different tasks are with respect to the distance $\|\theta_i - \theta_j\|$.

**Theorem 7.** *The difference in expected optimal value between two MDPs $\mathcal{M}_{\theta_i}, \mathcal{M}_{\theta_j} \in \mathcal{M}_\Theta$ is bounded by,*

$$|V_{\theta_i}^* - V_{\theta_j}^*| \leq \frac{\gamma}{(1-\gamma)^2} \|\theta_i - \theta_j\|_1. \tag{8}$$

*Proof.*

$$|V_{\theta_i}^*(s) - V_{\theta_j}^*(s)| = |\max_a Q_{\theta_i}^*(s,a) - \max_{a'} Q_{\theta_j}^*(s,a')|$$

$$\leq \max_a |Q_{\theta_i}^*(s,a) - Q_{\theta_j}^*(s,a)|$$

We can bound the RHS with

$$\sup_{s,a} |Q_{\theta_i}^*(s,a) - Q_{\theta_j}^*(s,a)| \leq \sup_{s,a} |r_{\theta_i}(s,a) - r_{\theta_j}(s,a)| + \gamma \sup_{s,a} |\mathbb{E}_{s' \sim T_{\theta_i}(\cdot|s,a)} V_{\theta_i}^*(s') - \mathbb{E}_{s'' \sim T_{\theta_j}(\cdot|s,a)} V_{\theta_j}^*(s'')|$$

All MDPs in $\mathcal{M}_\Theta$ have the same reward function, so the first term is 0.

$$\sup_{s,a} |Q_{\theta_i}^*(s,a) - Q_{\theta_j}^*(s,a)| \leq \gamma \sup_{s,a} |\mathbb{E}_{s' \sim T_{\theta_i}(\cdot|s,a)} V_{\theta_i}^*(s') - \mathbb{E}_{s'' \sim T_{\theta_j}(\cdot|s,a)} V_{\theta_j}^*(s'')|$$

$$= \gamma \sup_{s,a} \left| \mathbb{E}_{s' \sim T_{\theta_i}(\cdot|s,a)} \left[ V_{\theta_i}^*(s') - V_{\theta_j}^*(s') \right] + \mathbb{E}_{\substack{s'' \sim T_{\theta_j}(\cdot|s,a), \\ s' \sim T_{\theta_i}(\cdot|s,a)}} \left[ V_{\theta_j}^*(s') - V_{\theta_j}^*(s'') \right] \right|$$

$$\leq \gamma \sup_{s,a} \left| \mathbb{E}_{s' \sim T_{\theta_i}(\cdot|s,a)} \left[ V_{\theta_i}^*(s') - V_{\theta_j}^*(s') \right] \right| + \gamma \sup_{s,a} \left| \mathbb{E}_{\substack{s'' \sim T_{\theta_j}(\cdot|s,a), \\ s' \sim T_{\theta_i}(\cdot|s,a)}} \left[ V_{\theta_j}^*(s') - V_{\theta_j}^*(s'') \right] \right|$$

$$\leq \gamma \sup_{s,a} \left| \mathbb{E}_{s' \sim T_{\theta_i}(\cdot|s,a)} \left[ V_{\theta_i}^*(s') - V_{\theta_j}^*(s') \right] \right| + \frac{\gamma}{1-\gamma} \|\theta_i - \theta_j\|_1$$

$$\leq \gamma \max_s \left| V_{\theta_i}^*(s) - V_{\theta_j}^*(s) \right| + \frac{\gamma}{1-\gamma} \|\theta_i - \theta_j\|_1$$

$$= \gamma \max_s \left| \max_a Q_{\theta_i}^*(s,a) - \max_{a'} Q_{\theta_j}^*(s,a') \right| + \frac{\gamma}{1-\gamma} \|\theta_i - \theta_j\|_1$$

$$\leq \gamma \sup_{s,a} \left| Q_{\theta_i}^*(s,a) - Q_{\theta_j}^*(s,a) \right| + \frac{\gamma}{1-\gamma} \|\theta_i - \theta_j\|_1$$

Solving for $\sup_{s,a} \left| Q_{\theta_i}^*(s,a) - Q_{\theta_j}^*(s,a) \right|$,

$$\sup_{s,a} \left| Q_{\theta_i}^*(s,a) - Q_{\theta_j}^*(s,a) \right| \leq \frac{\gamma}{(1-\gamma)^2} \|\theta_i - \theta_j\|_1.$$

Plugging this back in,

$$|V_{\theta_i}^*(s) - V_{\theta_j}^*(s)| \leq \frac{\gamma}{(1-\gamma)^2} \|\theta_i - \theta_j\|_1.$$

$\square$

Both these results lend more intuition for casting the multi-task setting under the HiP-MDP formalism. The difference in the optimal performance between any two environments is controlled by the distance between the hidden parameters for corresponding environments. One can interpret the hidden parameter as a knob to allow precise changes across the tasks.

## B.2 EXPECTED ERROR BOUNDS

In MTRL, we are concerned with the performance over a family of tasks. The empirical risk is typically defined as follows for $T$ tasks (Maurer et al., 2016):

$$\epsilon_{avg}(\theta) = \frac{1}{T} \sum_{t=1}^{T} \mathbb{E}[\ell(f_t(h(w_t(X))), Y))]. \tag{9}$$

Consequently, we bound the expected loss over the family of environments $\mathcal{E}$ with respect to $\theta$. In particular, we are interested in the average approximation error and define it as the absolute model error averaged across all environments:

$$\epsilon_{avg}(\theta) = \frac{1}{|\mathcal{E}|} \sum_{i=1}^{\mathcal{E}} \left| V_{\hat{\theta}_i}^*(s) - V_{\theta_i}^*(s) \right|. \tag{10}$$

**Theorem 8.** *Given a family of environments $\mathcal{M}_\Theta$, each parameterized with an underlying true hidden parameter $\theta_1, \theta_2, \cdots, \theta_\mathcal{E}$, and let $\hat{\theta}_1, \hat{\theta}_2, \cdots, \hat{\theta}_\mathcal{E}$ be their respective approximations such that the average approximation error across all environments is bounded as follows:*

$$\epsilon_{avg}(\theta) \leq \frac{\epsilon\gamma}{(1-\gamma)^2}, \tag{11}$$

*where each environment's parameter $\theta_i$ is $\epsilon$-close to its approximation $\hat{\theta}_i$ i.e. $d(\hat{\theta}_i, \theta_i) \leq \epsilon$, where $d$ is the distance metric defined in Eq. 1.*

*Proof.* We here consider the approximation error averaged across all environments as follows:

$$\epsilon_{avg}(\theta) = \frac{1}{\mathcal{E}} \sum_{i=1}^{\mathcal{E}} \left| V_{\hat{\theta}_i}^*(s) - V_{\theta_i}^*(s) \right|$$

$$\epsilon_{avg}(\theta) = \frac{1}{\mathcal{E}} \sum_{i=1}^{\mathcal{E}} |\max_a Q_{\hat{\theta}_i}^*(s, a) - \max_{a'} Q_{\theta_i}^*(s, a')| $$
$$\leq \frac{1}{\mathcal{E}} \sum_{i=1}^{\mathcal{E}} \max_a |Q_{\hat{\theta}_i}^*(s, a) - Q_{\theta_i}^*(s, a)| \tag{12}$$

Let us consider an environment $\theta_i \in \mathcal{M}_\mathcal{E}$ for which we can bound the RHS with

$$\sup_{s,a} |Q_{\hat{\theta}_i}^*(s,a) - Q_{\theta_i}^*(s,a)| \leq \sup_{s,a} |r_{\hat{\theta}_i}(s,a) - r_{\theta_i}(s,a)| + \gamma \sup_{s,a} |\mathbb{E}_{s' \sim T_{\hat{\theta}_i}(\cdot|s,a)} V_{\hat{\theta}_i}^*(s') - \mathbb{E}_{s'' \sim T_{\theta_i}(\cdot|s,a)} V_{\theta_i}^*(s'')|$$

Considering the family of environments $\mathcal{M}_\mathcal{E}$ have the same reward function and is known, resulting in first term to be 0.

$$\sup_{s,a} \left| Q^*_{\hat{\theta}_i}(s,a) - Q^*_{\theta_i}(s,a) \right| \leq \gamma \sup_{s,a} \left| \mathbb{E}_{s' \sim T_{\hat{\theta}_i}(\cdot|s,a)} V^*_{\hat{\theta}_i}(s') - \mathbb{E}_{s'' \sim T_{\theta_i}(\cdot|s,a)} V^*_{\theta_i}(s'') \right|$$

$$= \gamma \sup_{s,a} \left| \mathbb{E}_{s' \sim T_{\hat{\theta}_i}(\cdot|s,a)} \left[ V^*_{\hat{\theta}_i}(s') - V^*_{\theta_i}(s') \right] + \mathbb{E}_{\substack{s'' \sim T_{\theta_i}(\cdot|s,a), \\ s' \sim T_{\hat{\theta}_i}(\cdot|s,a)}} \left[ V^*_{\theta_i}(s') - V^*_{\theta_i}(s'') \right] \right|$$

$$\leq \gamma \sup_{s,a} \left| \mathbb{E}_{s' \sim T_{\hat{\theta}_i}(\cdot|s,a)} \left[ V^*_{\hat{\theta}_i}(s') - V^*_{\theta_i}(s') \right] \right| + \gamma \sup_{s,a} \left| \mathbb{E}_{\substack{s'' \sim T_{\theta_i}(\cdot|s,a), \\ s' \sim T_{\hat{\theta}_i}(\cdot|s,a)}} \left[ V^*_{\theta_i}(s') - V^*_{\theta_i}(s'') \right] \right|$$

$$\leq \gamma \sup_{s,a} \left| \mathbb{E}_{s' \sim T_{\hat{\theta}_i}(\cdot|s,a)} \left[ V^*_{\hat{\theta}_i}(s') - V^*_{\theta_i}(s') \right] \right| + \frac{\gamma}{1-\gamma} |\hat{\theta}_i - \theta_i|$$

$$\leq \gamma \max_{s} \left| V^*_{\hat{\theta}_i}(s) - V^*_{\theta_i}(s) \right| + \frac{\gamma}{1-\gamma} |\hat{\theta}_i - \theta_i|$$

$$= \gamma \max_{s} \left| \max_{a} Q^*_{\hat{\theta}_i}(s,a) - \max_{a'} Q^*_{\theta_i}(s,a') \right| + \frac{\gamma}{1-\gamma} |\hat{\theta}_i - \theta_i|$$

$$\leq \gamma \sup_{s,a} \left| Q^*_{\hat{\theta}_i}(s,a) - Q^*_{\theta_i}(s,a) \right| + \frac{\gamma}{1-\gamma} |\hat{\theta}_i - \theta_i|$$

Solving for $\sup_{s,a} \left| Q^*_{\hat{\theta}_i}(s,a) - Q^*_{\theta_i}(s,a) \right|$,

$$\sup_{s,a} \left| Q^*_{\hat{\theta}_i}(s,a) - Q^*_{\theta_i}(s,a) \right| \leq \frac{\gamma}{(1-\gamma)^2} |\hat{\theta}_i - \theta_i| \tag{13}$$

Plugging Eq. 13 back in Eq. 12,

$$\epsilon_{avg}(\theta) \leq \frac{1}{\mathcal{E}} \sum_{i=1}^{\mathcal{E}} \frac{\gamma}{(1-\gamma)^2} |\hat{\theta}_i - \theta_i|$$

$$= \frac{\gamma}{\mathcal{E}(1-\gamma)^2} \left[ |\hat{\theta}_{i=1} - \theta_{i=1}| + |\hat{\theta}_{i=2} - \theta_{i=2}| + \cdots + |\hat{\theta}_{i=\mathcal{E}} - \theta_{i=\mathcal{E}}| \right]$$

We now consider that the distance between the approximated $\hat{\theta}_i$ and the underlying hidden parameter $\theta_i \in \mathcal{M}_{\mathcal{E}}$ is defined as in Eq. 1, such that: $d(\hat{\theta}_i, \theta_i) \leq \epsilon_\theta$

Plugging this back concludes the proof,

$$\epsilon_{avg}(\theta) \leq \frac{\gamma \epsilon_\theta}{(1-\gamma)^2}.$$

$\square$

It is interesting to note that the average approximation error across all environments is independent of the number of environments and primarily governed by the error in approximating the hidden parameter $\theta$ for each environment.

## C  ADDITIONAL RESULTS AND PROOFS FOR HIP-BMDP RESULTS

We first compute $L^\infty$ norm bounds for $Q$ error under approximate abstractions and transfer bounds.

**Theorem 9** ($Q$ error). *Given an MDP $\bar{\mathcal{M}}_{\hat{\theta}}$ built on a $(\epsilon_R, \epsilon_T, \epsilon_\theta)$-approximate bisimulation abstraction of an instance of a HiP-BMDP $\mathcal{M}_\theta$, we denote the evaluation of the optimal $Q$ function of $\bar{\mathcal{M}}_{\hat{\theta}}$ on $\mathcal{M}$ as $[Q^*_{\bar{\mathcal{M}}_{\hat{\theta}}}]_{\mathcal{M}_\theta}$. The value difference with respect to the optimal $Q^*_{\mathcal{M}}$ is upper bounded by*

$$\left\| Q^*_{\mathcal{M}_\theta} - [Q^*_{\bar{\mathcal{M}}_{\hat{\theta}}}]_{\mathcal{M}_\theta} \right\|_\infty \leq \epsilon_R + \gamma(\epsilon_T + \epsilon_\theta) \frac{R_{max}}{2(1-\gamma)}.$$

*Proof.* In the HiP-BMDP setting, we have a global encoder $\phi$ over all tasks, but the difference in transition distribution also includes $\theta$. The reward functions are the same across tasks, so there is

no change to $\epsilon_R$. However, we now must incorporate difference in dynamics in $\epsilon_T$. Assuming we have two environments with hidden parameters $\theta_i, \theta_j \in \Theta$, we can compute $\epsilon_T^{\theta_i, \theta_j}$ across those two environments by joining them into a super-MDP:

$$
\begin{aligned}
\epsilon_T^{\theta_i, \theta_j} &= \sup_{\substack{a \in \mathcal{A}, \\ x_1, x_2 \in \mathcal{X}, \phi(x_1) = \phi(x_2)}} \left\| \Phi T_{\theta_i}(x_1, a) - \Phi T_{\theta_j}(x_2, a) \right\|_1 \\
&\leq \sup_{\substack{a \in \mathcal{A}, \\ x_1, x_2 \in \mathcal{X}, \phi(x_1) = \phi(x_2)}} \left( \left\| \Phi T_{\theta_i}(x_1, a) - \Phi T_{\theta_i}(x_2, a) \right\|_1 + \left\| \Phi T_{\theta_i}(x_2, a) - \Phi T_{\theta_j}(x_2, a) \right\|_1 \right) \\
&\leq \sup_{\substack{a \in \mathcal{A}, \\ x_1, x_2 \in \mathcal{X}, \phi(x_1) = \phi(x_2)}} \left\| \Phi T_{\theta_i}(x_1, a) - \Phi T_{\theta_i}(x_2, a) \right\|_1 + \sup_{\substack{a \in \mathcal{A}, \\ x_1, x_2 \in \mathcal{X}, \phi(x_1) = \phi(x_2)}} \left\| \Phi T_{\theta_i}(x_2, a) - \Phi T_{\theta_j}(x_2, a) \right\|_1 \\
&= \epsilon_T^{\theta_i} + \| \theta_i - \theta_j \|_1
\end{aligned}
$$

This result is intuitive in that with a shared encoder learning a per-task bisimulation relation, the distance between bisimilar states from another task depends on the change in transition distribution between those two tasks. We can now extend the single-task bisimulation bound (Theorem 5) to the HiP-BMDP setting by denoting approximation error of $\theta$ as $\| \theta - \hat{\theta} \|_1 < \epsilon_\theta$. $\qquad\square$

**Theorem 4.** *For any $\phi$ which defines an $(\epsilon_R, \epsilon_T, \epsilon_\theta)$-approximate bisimulation abstraction on a HiP-BMDP family $\mathcal{M}_\Theta$, we define the empirical measurement of $Q^*_{\bar{\mathcal{M}}_{\hat{\theta}}}$ over $D$ to be $Q^*_{\bar{\mathcal{M}}^D_{\hat{\theta}}}$. Then, with probability $\geq 1 - \delta$,*

$$
\left\| Q^*_{\mathcal{M}_\theta} - [Q^*_{\bar{\mathcal{M}}^D_{\hat{\theta}}}] \mathcal{M}_\theta \right\|_\infty \leq \epsilon_R + \gamma(\epsilon_T + \epsilon_\theta) \frac{R_{max}}{2(1-\gamma)} + \frac{R_{max}}{(1-\gamma)^2} \sqrt{\frac{1}{2 n_\phi(D)} \log \frac{2|\phi(\mathcal{X})||\mathcal{A}|}{\delta}}. \quad (14)
$$

*Proof.*

$$
\begin{aligned}
\left\| Q^*_{\mathcal{M}_\theta} - [Q^*_{\bar{\mathcal{M}}^D_{\hat{\theta}}}] \mathcal{M}_\theta \right\|_\infty &\leq \left\| Q^*_{\mathcal{M}_\theta} - [Q^*_{\bar{\mathcal{M}}_{\hat{\theta}}}] \mathcal{M}_\theta \right\|_\infty + \left\| [Q^*_{\bar{\mathcal{M}}_{\hat{\theta}}}] \mathcal{M}_\theta - [Q^*_{\bar{\mathcal{M}}^D_{\hat{\theta}}}] \mathcal{M}_\theta \right\|_\infty \\
&= \left\| Q^*_{\mathcal{M}_\theta} - [Q^*_{\bar{\mathcal{M}}_{\hat{\theta}}}] \mathcal{M}_\theta \right\|_\infty + \left\| Q^*_{\bar{\mathcal{M}}_{\hat{\theta}}} - Q^*_{\bar{\mathcal{M}}^D_{\hat{\theta}}} \right\|_\infty
\end{aligned}
$$

The first term is solved by Theorem 2, so we only need to solve the second term using McDiarmid's inequality and the knowledge that the value function of a bisimulation representation is $\frac{1}{1-\gamma}$-Lipschitz from Theorem 1.

First, we write this difference to be a deviation from an expectation in order to apply the concentration inequality.

$$
\begin{aligned}
\| Q^*_{\bar{\mathcal{M}}_{\hat{\theta}}} - Q^*_{\bar{\mathcal{M}}^D_{\hat{\theta}}} \|_\infty &= \| Q^*_{\bar{\mathcal{M}}_{\hat{\theta}}} - \mathcal{T}^\phi_D Q^*_{\bar{\mathcal{M}}_{\hat{\theta}}} + \mathcal{T}^\phi_D Q^*_{\bar{\mathcal{M}}_{\hat{\theta}}} - \mathcal{T}^\phi_D Q^*_{\bar{\mathcal{M}}^D_{\hat{\theta}}} \|_\infty \\
&\leq \| Q^*_{\bar{\mathcal{M}}_{\hat{\theta}}} - \mathcal{T}^\phi_D Q^*_{\bar{\mathcal{M}}_{\hat{\theta}}} \|_\infty + \gamma \| Q^*_{\bar{\mathcal{M}}_{\hat{\theta}}} - Q^*_{\bar{\mathcal{M}}^D_{\hat{\theta}}} \|_\infty \\
&\leq \frac{1}{1-\gamma} \| \mathcal{T}^\phi_D Q^*_{\bar{\mathcal{M}}_{\hat{\theta}}} - \mathcal{T}^\phi Q^*_{\bar{\mathcal{M}}_{\hat{\theta}}} \|_\infty
\end{aligned}
$$

Now we can apply McDiarmid's inequality,

$$
\mathbb{P}_D \left[ |Q^*_{\bar{\mathcal{M}}_{\hat{\theta}}} - Q^*_{\bar{\mathcal{M}}^D_{\hat{\theta}}}| \geq t \right] \leq 2 \exp \left( - \frac{2 t^2 |D_{\phi(x), a}|}{R_{max}^2 / (1-\gamma)^2} \right).
$$

Solve for the $t$ that makes this inequality hold for all $(\phi(x), a) \in \mathcal{X} \times \mathcal{A}$ with a union bound over all $|\phi(\mathcal{X})||\mathcal{A}|$ abstract states,

$$
t > \frac{R_{max}}{1-\gamma} \sqrt{\frac{1}{2 n_\phi(D)} \log \frac{2|\phi(\mathcal{X})||\mathcal{A}|}{\delta}}.
$$

Combine to get

$$
\left\| Q^*_{\mathcal{M}_\theta} - [Q^*_{\bar{\mathcal{M}}^D_{\hat{\theta}}}] \mathcal{M}_\theta \right\|_\infty \leq \epsilon_R + \gamma(\epsilon_T + \epsilon_\theta) \frac{R_{max}}{2(1-\gamma)} + \frac{R_{max}}{(1-\gamma)^2} \sqrt{\frac{1}{2 n_\phi(D)} \log \frac{2|\phi(\mathcal{X})||\mathcal{A}|}{\delta}}.
$$

$\qquad\square$

---

**Algorithm 1 HiP-BMDP training for the Multi-task RL setting.**

---

**Require:** Along with DeepMDP components (Actor, Critic, Dynamics Model ($M$)), an additional
environment encoder $\psi$ to generated task-specific $\theta$ parameters.

1: **for** each timestep $t = 1..T$ **do**
2:     **for** each $\mathcal{T}_i$ **do**
3:        $a_t^i \sim \pi^i(\cdot|s_t^i)$
4:        $s_t'^i \sim p^i(\cdot|s_t^i, a_t^i)$
5:        $\mathcal{D} \leftarrow \mathcal{D} \cup (s_t^i, a_t^i, r(s_t^i, a_t^i), s_t'^i), i$
6:        UPDATECRITIC($\mathcal{D}, i$) (uses data only from $i^{th}$ task)
7:        UPDATEACTOR($\mathcal{D}, i$) (uses data only from $i^{th}$ task)
8:        UPDATEUSINGHIP-BMDPLOSS($\mathcal{D}, i$)
9:     **end for**
10: **end for**

---

**Algorithm 2 UpdateModelUsingHip-BMDPLoss**

---

**Require:** Batches of data for the different tasks $\{\mathcal{T}_i\}_{i=1...T}$ sampled from the Replay Buffer $\mathcal{D}$,
learning rates $\alpha_1$ and $\alpha_2$, index of the current task $i$, Transition Model $M$, environment encoder
$\psi$.

1: **for** each batch of dataset $t = 1..T, t \neq i$ **do**
2:     Compute $\mathcal{L}(\psi, M) = \mathcal{L}^i(\psi, M, i, t)$ using Equation (3)
3:     $\psi \leftarrow \psi - \alpha_1 \nabla_\theta \sum_i \mathcal{L}$
4:     $M \leftarrow M - \alpha_2 \nabla_\theta \sum_i \mathcal{L}$
5: **end for**

---

# D   ADDITIONAL IMPLEMENTATION DETAILS

In Figure 8, we show the variation in the left foot of the walker.

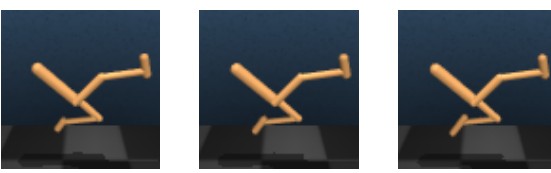

Figure 8: Variation in Walker (V1) across different tasks.

**MTRL Algorithm**     The Multitask RL algorithm for the HiP-BMDP setting can be found in Algorithm 1. We take the DeepMDP baseline Gelada et al. (2019) and incorporate our HiP-BMDP objective (text shown in red color)

**Meta-RL Algorithm**     The meta-RL algorithm for the HiP-MDP setting can be found in Algorithm 3. We take the PEARL algorithm (Rakelly et al., 2019) and incorporate our HiP-MDP objective (text shown in red color)

## D.1   BASELINES

For PCGrad, the authors recommend projecting the gradient with respect to all previous tasks. In practice, that leads to very poor training. Instead, we observe that it is better to project the gradients with respect to any one task (randomly selected per update). We use this scheme in all the experiments. For GradNorm, we observe that the learned weights $w_i$ (for weighing per-task loss) can become negative for some tasks (which means the model tries to unlearn those tasks). In practice, we clamp the $w_i$ values to not become smaller than a threshold.

## D.2   HYPER PARAMETERS

### D.2.1   MTRL ALGORITHM

All the hyper parameters (for MTRL algorithm) are listed in Table 1.

| Parameter name | Value |
|---|---|
| Actor learning rate | $10^{-3}$ |
| Actor update frequency | 2 |
| Actor log stddev bounds | $[-10, 2]$ |
| Alpha learning rate | $10^{-4}$ |
| Batch size | 128 |
| Critic learning rate | $10^{-3}$ |
| Critic target update frequency | 2 |
| Critic Q-function soft-update rate $\tau_Q$ | 0.01 |
| Critic encoder soft-update rate $\tau_{enc}$ | 0.05 |
| Discount $\gamma$ | 0.99 |
| Decoder learning rate | $10^{-3}$ |
| Critic update frequency | 1 |
| Encoder feature dimension | 100 |
| Encoder learning rate | $10^{-3}$ |
| Number of encoder layers | 4 |
| Number of filters in encoder | 32 |
| Hidden Dimension | 1024 |
| Replay buffer capacity | 1000000 |
| Optimizer | Adam |
| Temperature Adam's $\beta_1$ | 0.5 |
| Init temperature | 0.1 |
| Number of embeddings in $\psi$ | 10 |
| Embedding dimension for $\psi$ | 100 |
| Learning rate for $\psi$ | $10^{-3}$ |
| $\alpha_\psi$ (ie $\alpha$ for $\psi$) for Finger-Spin-V0 | 0.1 |
| $\alpha_\psi$ (ie $\alpha$ for $\psi$) for Cheetah-Run-V0 | 0.1 |
| $\alpha_\psi$ (ie $\alpha$ for $\psi$) for Walker-Run-V0 | 1.0 |
| $\alpha_\psi$ (ie $\alpha$ for $\psi$) for Walker-Run-V1 | 0.01 |
| $\alpha_{distal}$ (ie $\alpha$ for Distral) for Finger-Spin-V0 | 0.05 |
| $\alpha_{distal}$ (ie $\alpha$ for Distral) for Cheetah-Run-V0 | 0.01 |
| $\alpha_{distal}$ (ie $\alpha$ for Distral) for Walker-Run-V0 | 0.05 |
| $\alpha_{distal}$ (ie $\alpha$ for Distral) for Walker-Run-V1 | 0.01 |
| $\beta_{distal}$ (ie $\beta$ for Distral) | 1.0 |

Table 1: A complete overview of used hyper parameters.

### D.2.2   METARL ALGORITHM

For MetaRL, we use the same hyperparameters as used by PEARL Rakelly et al. (2019). We set $\alpha_\phi = 0.01$ for all environments, other than *Walker-Stand* environments where $\alpha_\phi = 0.001$.

## E   ADDITIONAL RESULTS

Along with the environments described in 4, we considered the following additional environments:

1. `Walker-Stand-V0`: `Walker-Stand` task where the friction coefficient, between the ground and the walker's leg, varies across different environments.
2. `Walker-Walk-V0`: `Walker-Walk` task where the friction coefficient, between the ground and the walker's leg, varies across different environments.
3. `Walker-Stand-V1`: `Walker-Stand` task where the size of left-foot of the walker varies across different environments.

---

**Algorithm 3 HiP-MDP training for the meta-RL setting.**

---

**Require:** Batch of training tasks $\{\mathcal{T}_i\}_{i=1...T}$ from $p(\mathcal{T})$, learning rates $\alpha_1, \alpha_2, \alpha_3, \alpha_\phi$
 1: Initialize replay buffers $\mathcal{B}^i$ for each training task
 2: **while** not done **do**
 3:     **for** each $\mathcal{T}_i$ **do**
 4:        Initialize context $C^i = \{\}$
 5:        **for** $k = 1, \ldots, K$ **do**
 6:           Sample $\mathbf{z} \sim q_\phi(\mathbf{z}|C^i)$
 7:           Gather data from $\pi_\theta(\mathbf{a}|\mathbf{s}, \mathbf{z})$ and add to $\mathcal{B}^i$
 8:           Update $C^i = \{(\mathbf{s}_j, \mathbf{a}_j, \mathbf{s}'_j, r_j)\}_{j:1...N} \sim \mathcal{B}^i$
 9:        **end for**
10:     **end for**
11:     **for** step in training steps **do**
12:        **for** each $\mathcal{T}_i$ **do**
13:           Sample context $C^i \sim \mathcal{S}_c(\mathcal{B}^i)$ and RL batch $b^i \sim \mathcal{B}^i$
14:           Sample $\mathbf{z} \sim q_\phi(\mathbf{z}|C^i)$
15:           $\mathcal{L}^i_{actor} = \mathcal{L}_{actor}(b^i, \mathbf{z})$
16:           $\mathcal{L}^i_{critic} = \mathcal{L}_{critic}(b^i, \mathbf{z})$
17:           $\mathcal{L}^i_{KL} = \beta D_{\text{KL}}(q(\mathbf{z}|C^i)||r(\mathbf{z}))$
18:           Sample a RL batch $b^j$ from any other task j
19:           Compute $\mathcal{L}^i_{BiSim} = \mathcal{L}^i(q, T, i, j)$ using the equation 3
20:        **end for**
21:        $\phi \leftarrow \phi - \alpha_1 \nabla_\phi \sum_i \left(\mathcal{L}^i_{critic} + \mathcal{L}^i_{KL} + \alpha_\phi \times \mathcal{L}^i_{BiSim}\right)$
22:        $\theta_\pi \leftarrow \theta_\pi - \alpha_2 \nabla_\theta \sum_i \mathcal{L}^i_{actor}$
23:        $\theta_Q \leftarrow \theta_Q - \alpha_3 \nabla_\theta \sum_i \mathcal{L}^i_{critic}$
24:     **end for**
25: **end while**

---

4. `Walker-Walk-V1`: `Walker-Walk` task where the size of left-foot of the walker varies across different environments.

### E.1 MULTI-TASK SETTING

In Figure 10, we observe that the HiP-BMDP method consistently outperforms other baselines when evaluated on the interpolation environments (zero-shot transfer). As noted previously, the effectiveness of our proposed model can not be attributed to task-embeddings alone as HiP-BMDP-nobisim model uses the same architecture as the HiP-BMDP model but does not include the task bisimulation metric loss. We hypothesise that the Distral-Ensemble baseline behaves poorly because it cannot leverage a shared global dynamics model.

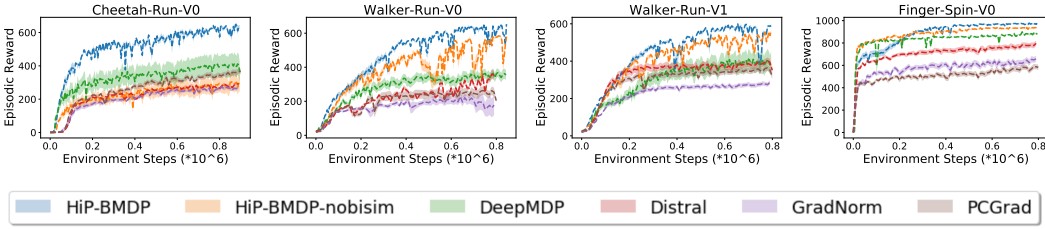

Figure 9: Multi-Task Setting. Performance on the training tasks. Note that the environment steps (on the x-axis) denote the environment steps for each task. Since we are training over four environments, the actual number of steps is approximately 3.2 million.

### E.2 META-RL SETTING

We provide the Meta-RL results for the additional environments. Recall that we extend the PEARL algorithm (Rakelly et al., 2019) by training the *inference network* $q_\phi(\mathbf{z}|\mathbf{c})$ with our additional HiP-BMDP loss. The algorithm pseudocode can be found in Appendix D. In Figure 13, we show the

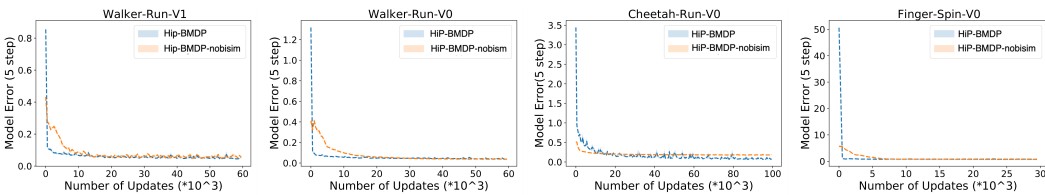

Figure 10: Zero-shot generalization performance on the interpolation tasks in the MTRL setting. HiP-BMDP (ours) consistently outperforms other baselines. 10 seeds, 1 standard error shaded.

Figure 11: Average per-step model error (in latent space) after unrolling the transition model for 5 steps.

results on the interpolation setup and in Figure 14, we show the results on the extrapolation setup. In some environments (eg Walker-Walk-V1), the proposed approach (blue) converges faster to a threshold reward (green) than the baseline. In the other environments, the gains are quite small.

### E.3   Evaluating the Universal Transition Model.

We investigate how well the transition model performs in an unseen environment by only adapting the task parameter $\theta$. We instantiate a new MDP, sampled from the family of MDPs, and use a behavior policy to collect transitions. These transitions are used to update only the $\theta$ parameter, and the transition model is evaluated by unrolling the transition model for $k$-steps. In Figures 11 and 12, we report the average, per-step model error in latent space, averaged over 10 environments over 5 and 100 steps respectively. While we expect both the proposed setup and baseline setups to adapt to the new environment, we expect the proposed setup to adapt faster because of the exploitation of underlying structure. We indeed observe that for both 5 step and 100 step unrolls, the proposed *HiP-BMDP* model adapts much faster than the baseline *HiP-BMDP-nobisim* (Figures 11 and 12)

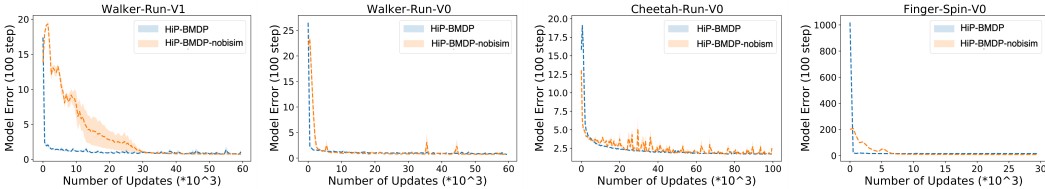

Figure 12: Average per-step model error (in latent space) after unrolling the transition model for 100 steps.

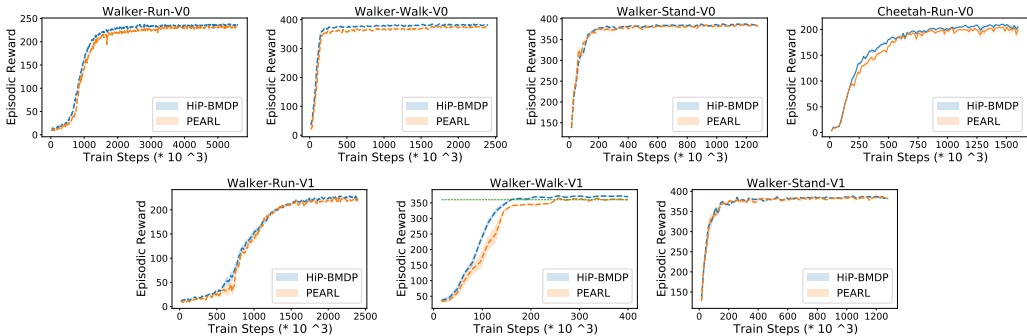

Figure 13: Few-shot generalization performance on the interpolation tasks on Walker-Run-V0, Walker-Walk-V0 Walker-Stand-V0, Cheetah-Run-V0 (top row), Walker-Run-V1, Walker-Walk-V1 and Walker-Stand-V1 (bottom row) respectively. We note that for the Walker-Walk-V1, the proposed approach (blue) converges faster to a threshold reward (green) than the baseline. In other environments, the gains are quite small.

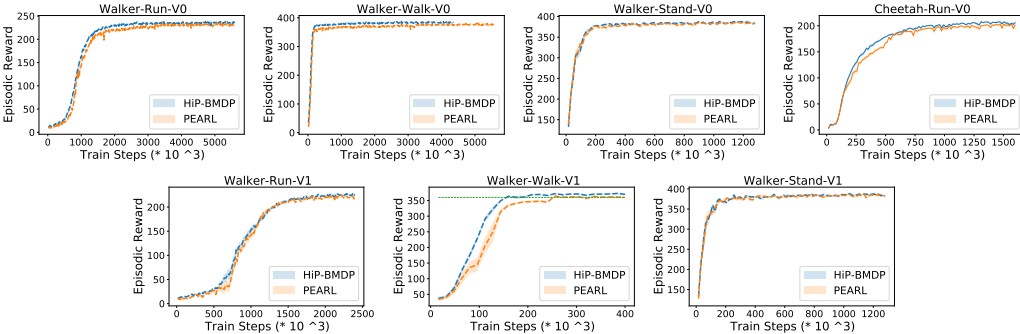

Figure 14: Few-shot generalization performance on the extrapolation tasks on Walker-Run-V0, Walker-Walk-V0, Walker-Stand-V0, Cheetah-Run-V0 (top row), Walker-Run-V1, Walker-Walk-V1 and Walker-Stand-V1 (bottom row) respectively. We note that for the Walker-Walk-V1, the proposed approach (blue) converges faster to a threshold reward (green) than the baseline. In other environments, the gains are quite small.

