# OpenReview forum: "Learning Robust State Abstractions for Hidden-Parameter Block MDPs"
_ICLR.cc/2021/Conference — ICLR 2021 Poster_

### Official Review · AnonReviewer3 · 2020-10-24
**Interesting paper which requires some improvements/clarifications**

**Rating:** 7
**Confidence:** 3

**Review:**

### Summary

The authors combine hidden-parameter MDPs and state abstractions to model multi-task problems with dynamics parameterized by some latent variables and where the agent receives high-dimensional observations whose corresponding low-dimensional states, on which dynamics are defined, are unobserved (as in block MDPs). They provide both a theoretical analysis and an empirical evaluation of the proposed method, showing that it performs better than competitive baselines on complex continuous control domains.

### Pros

- The idea of combining HiP-MDPs with state abstractions seems interesting and relevant. One of the limitations of HiP-MDPs is indeed that they do not easily handle high-dimensional observations and the proposed approach overcomes this limitation.
- The method seems also quite general, in the sense that it can be applied in multi-task/meta-RL settings and combined with existing algorithms.
- Experiments are conducted on complex tasks and show convincing results.

### Cons

- I found the paper quite hard to read in its core parts (e.g., Sec. 3). This is in part due to a complex/confused notation, which overall made it hard for me to go through the theoretical part (and proofs). See detailed comments below.
- The theoretical results focus on assessing the value-function errors for fixed abstractions/dynamics-structure rather than on the errors in learning the abstractions/structure themselves (and learning these components seems to be one of the primary concerns here).
- The requirement that task IDs are known could be a potential limitation for applying this method

### Detailed comments

1. Are the environment labels/ids used only for communicating to the learner that the task changed or could they provide more information? For instance, if we face the same task twice in two non-consecutive episodes, are the corresponding labels equal?

2. What makes these state abstractions "robust" (as in the title)? Is there any particular theoretical or empirical evidence that justifies this term?

3. Existing analyses of multi-task settings, including the one of Brunskill and Li [1] and others [2,3,4], provide guarantees for task-structure (e.g., dynamics models) learned from data, while here the focus seems to be on guarantees for arbitrary abstractions/structures as a function of their errors, without considering how they are learned. This makes it difficult to interpret the results in the context of the proposed approach

4. Regarding the statement: "sample complexity bounds that depend on the aggregate number of samples across tasks, rather than the number of tasks, a significant improvement over prior work", the bounds derived in [4] depend only on a number of abstract "transition templates" that they define, while those derived in [3] depend on the minimum between the number of tasks and the number of states.

5. To be clear, the notation \| x - y \|_1 is used for the Wasserstein distance and not l1-norm, right? In such a case, the sentence "We omit d but use d(x, y) = \|x − y\|_1 in our setting" is quite confusing.

6. The definitions of the terms \epsilon_R, \epsilon_T, \epsilon_\theta could be moved to the theoretical analysis since they are introduced early and never used before Sec 3.3

7. Sec. 3.2: the definition of \phi maps S to Z. Isn't it X to Z?

8. In Sec. 3.2, the 2-Wasserstein distance is used, while the 1-Wasserstein distance was introduced earlier. Which one is used?

9. In Eq. 3, it was not clear to me why the "model learning error" term uses the MSE for two different environment labels (isn't one enough)?

10. What is exactly the "evaluation of the optimal Q function of \bar{M} on M" in Theorem 2? Do you mean that we take the optimal policy of \bar{M} and test it on M?

11. [if the above comment is correct] Theorem 2 was slightly weird to me: it takes the optimal policy of \bar{M}, evaluates it on M_{\theta_i}, and check how much the corresponding Q function differs from the optimal Q function of M_{\theta_j}. If we wanted to figure out the transfer error of the optimal policy of \bar{M} on M_{\theta_j}, shouldn't we compare the optimal Q function of M_{\theta_j} with the Q function of the policy on the same MDP (rather than on M_{\theta_i})?

12. How exactly is the "empirical measurement of Q^*_{\bar{M}} over D" computed?

13. In Th. 4, the last term bounds the concentration of expectations of Q functions (which are bounded by rmax/(1-\gamma). Shouldn't there be a multiplicative rmax in front of the sqrt?

14. Page 6: "We compare the our method" -> "We compare our method"

15. In the multi-task experiments, it was not clear what method was used in combination with HiP-BMDPs to learn policies (though it is in appendix).

16. How many adaptation steps for "few-shot generalization" were used in Figure 5?

17. Below Figure 5: "and use a behavior policy to collect transitions." What is the behavior policy exactly and how was it chosen?

Some minor comments/questions:

- Page 1: "This additional structure gives us better sample efficiency, both theoretically and empirically." Better than what algorithm?
- Page 2: "don't" -> "do not"
- Page 2: "which naturally leads to a gradient-based representation learning algorithm." Why is it natural to use a gradient-based algorithm in this setting?
- Background: you could mention that the finite-MDP assumption is only for deriving the theoretical results, while the method can be applied in continuous domains (right?)
- Page 2: "a unobservable "-> "an unobservable"
- Page 3: "where l2 distance corresponds to d" Why? d isn't the l2 distance, right?
- Sec. 3.1: \phi in \epsilon was never defined.

### Overall comment

Overall I believe that the paper is interesting and the results significant. However, at the present time, I have too many doubts to vote for acceptance. I will be happy to increase my score after the authors have clarified them.

### Update

I have increased my score to 7 after reading the authors' response and the updated paper.

### References

[1] Brunskill, Emma, and Lihong Li. "Sample complexity of multi-task reinforcement learning." UAI (2013).
[2] Liu, Yao, Zhaohan Guo, and Emma Brunskill. "Pac continuous state online multitask reinforcement learning with identification." AAMAS (2016).
[3] Tirinzoni, Andrea, Riccardo Poiani, and Marcello Restelli. "Sequential transfer in reinforcement learning with a generative model." ICML (2020).
[4] Sun, Yanchao, Xiangyu Yin, and Furong Huang. "TempLe: Learning Template of Transitions for Sample Efficient Multi-task RL." arXiv (2020).

---

> ### Author Response · Authors · 2020-11-20
> **Thank you for your very helpful review**
>
> Thank you for your thoughtful response and detailed review. To address your main concerns: we did not focus on the errors in learning the abstractions and structure themselves because these are supervised learning problems with well understood bounds. Assuming that we can get low error with these, and measuring what that error is, we want to understand how this affects the downstream control problem. The requirement that task IDs are known is truly a potential limitation, as we also bring up in our discussion (Section 6). As we note there, we can incorporate a system identification procedure at training time to replace this assumption. Further, we note that most MTRL literature including the baselines we compare against, similarly assume this task id knowledge. Finally, we have improved the readability of the paper based on your comments (and fixed some typos and errors!), and also individually address each point below.
>
> 1. Are the environment labels/ids used only for communicating to the learner that the task changed or could they provide more information? For instance, if we face the same task twice in two non-consecutive episodes, are the corresponding labels equal?
> A: The environment labels are tied to the environment throughout training. Therefore, yes, if we face the same task twice in non-consecutive episodes, the corresponding labels are equal.
>
> 2. What makes these state abstractions "robust" (as in the title)? Is there any particular theoretical or empirical evidence that justifies this term?
> A: We chose to call these state abstractions “robust” due to two reasons: 1) Our theoretical analysis of value bounds (Sec 3.3) quantifies the error in transferring a policy learned on task i to a task j is primarily governed by the distance in the task-space defined by the hidden parameter \theta. This translates to the robustness of a policy learned in task i when evaluated in a task j, 2) Our empirical analysis of learning state abstractions for HiP-MDPs further corroborate our claims on how “robust” the learned representations are across a family of tasks (Sec. 4).
>
> 3. Existing analyses of multi-task settings, including the one of Brunskill and Li [1] and others [2,3,4], provide guarantees for task-structure (e.g., dynamics models) learned from data, while here the focus seems to be on guarantees for arbitrary abstractions/structures as a function of their errors, without considering how they are learned. This makes it difficult to interpret the results in the context of the proposed approach.
> A: Yes, that is correct. One drawback of existing guarantees is that they are tied to a specific algorithm and do not handle deep learning family of approaches. However, the guarantees presented in the proposed approach are learning approach agnostic for any abstractions which define an approximate bisimulation on a HiP-BMDP family.
>
> 4. Regarding the statement: "sample complexity bounds that depend on the aggregate number of samples across tasks, rather than the number of tasks, a significant improvement over prior work", the bounds derived in [4] depend only on a number of abstract "transition templates" that they define, while those derived in [3] depend on the minimum between the number of tasks and the number of states.
> A: Thank you for these references, we will add them to related work. We will rephrase our claim in the abstract. However, we argue that our method is superior to those in [3, 4] --  [4] only works with discrete state spaces, and does not scale well as the abstraction they learn for each state-action pair is of dimension |S|+1. [3] relies on a generative model to query any possible state-action pair -- an unreasonable assumption in most environments.  Further, since [3] depends on the minimum between number of tasks and number of states, and in our setting states always vastly outnumber the number of available tasks, we argue that [3] still depends on number of tasks.
>
> 5. To be clear, the notation | x - y |_1 is used for the Wasserstein distance and not l1-norm, right? In such a case, the sentence "We omit d but use d(x, y) = |x − y|_1 in our setting" is quite confusing.
> A: This notation refers to l1-norm in the constructed task embedding space, which corresponds to Wasserstein distance between probability distributions across tasks.
>
> 6. The definitions of the terms \epsilon_R, \epsilon_T, \epsilon_\theta could be moved to the theoretical analysis since they are introduced early and never used before Sec 3.3.
> A: Moved!

---

> > ### Author Response · Authors · 2020-11-20
> > **Response Cont.**
> >
> > 7. Sec. 3.2: the definition of \phi maps S to Z. Isn't it X to Z?
> > A: Yes, thank you for the catch. Fixed!
> >
> > 8. In Sec. 3.2, the 2-Wasserstein distance is used, while the 1-Wasserstein distance was introduced earlier. Which one is used?
> > A: We use the 2-Wasserstein distance because the W_2 metric has a convenient closed form. We have corrected the theta objective equation to use W_2 to be consistent.
> >
> > 9. In Eq. 3, it was not clear to me why the "model learning error" term uses the MSE for two different environment labels (isn't one enough)?
> > A: Yes. The two terms here just reflect the two transitions used to compute the bisimulation distance for training the task representation.
> >
> > 10. What is exactly the "evaluation of the optimal Q function of \bar{M} on M" in Theorem 2? Do you mean that we take the optimal policy of \bar{M} and test it on M?
> > A: Yes.
> >
> > 11. [if the above comment is correct] Theorem 2 was slightly weird to me: it takes the optimal policy of \bar{M}, evaluates it on M_{\theta_i}, and check how much the corresponding Q function differs from the optimal Q function of M_{\theta_j}. If we wanted to figure out the transfer error of the optimal policy of \bar{M} on M_{\theta_j}, shouldn't we compare the optimal Q function of M_{\theta_j} with the Q function of the policy on the same MDP (rather than on M_{\theta_i})?
> > A: Apologies for the confusion -- this was a typo in the theorem! Thank you for the catch. You’re correct, this is the optimal policy of \bar{M} applied to M_{\theta_j}. We have fixed it in the PDF.
> >
> > 12. How exactly is the "empirical measurement of Q^*_{\bar{M}} over D" computed?
> > A: This is the optimal value function of the MDP constructed from our learned state abstraction as computed from data D. This is computed with tabular value iteration with data in D.
> >
> > 13. In Th. 4, the last term bounds the concentration of expectations of Q functions (which are bounded by rmax/(1-\gamma). Shouldn't there be a multiplicative rmax in front of the sqrt?
> > A: You are correct. Fixed.
> >
> > 14. Page 6: "We compare the our method" -> "We compare our method"
> > A: fixed
> >
> > 15. In the multi-task experiments, it was not clear what method was used in combination with HiP-BMDPs to learn policies (though it is in appendix).
> > A: We have added a paragraph to the Background section to detail that we are using Soft Actor-Critic for downstream evaluation.
> >
> > 16. How many adaptation steps for "few-shot generalization" were used in Figure 5?
> > A: 100 steps, as used in PEARL.
> >
> > 17. Below Figure 5: "and use a behavior policy to collect transitions." What is the behavior policy exactly and how was it chosen?
> > A: The behavior policy is an expert policy trained on this task, because we are interested in how well the model generalizes on a data distribution collected by an optimal policy, which is a very different prediction problem (and likely more difficult) than from data generated by a random policy.
> >
> > Some minor comments/questions:
> > a. Page 1: "This additional structure gives us better sample efficiency, both theoretically and empirically." Better than what algorithm?
> > A: Better than baselines and theoretical results of other works presented in the related work, such as Brunskill et al. 2013. Added this detail to this sentence in the updated draft.
> >
> > b. Page 2: "don't" -> "do not" fixed
> >
> > c. Page 2: "which naturally leads to a gradient-based representation learning algorithm." Why is it natural to use a gradient-based algorithm in this setting?
> > A: Our wording was meant to imply that this representation learning objective can be optimized with a gradient-based algorithm.
> >
> > c. Background: you could mention that the finite-MDP assumption is only for deriving the theoretical results, while the method can be applied in continuous domains (right?)
> > A: correct, added a footnote to Background to note this.
> >
> > d. Page 2: "a unobservable "-> "an unobservable" fixed
> >
> > e. Page 3: "where l2 distance corresponds to d" Why? d isn't the l2 distance, right?
> > A: This was meant to give intuition that if you have a state space (or learned representation) where d is the distance metric, you have a Lipschitz MDP. We have simplified this to say V* is Lipschitz with respect to d.
> >
> > f. Sec. 3.1: \phi in \epsilon was never defined.
> > A: Apologies -- the definition was accidentally moved to the appendix. Adding back.

---

> > > ### Comment · AnonReviewer3 · 2020-11-24
> > > **Response to authors' feedback**
> > >
> > > Dear authors,
> > >
> > > Thank you for your very detailed response and for updating the manuscript. That addresses all my comments. I have increased my score accordingly and I now vote for acceptance.

---

### Official Review · AnonReviewer4 · 2020-10-25
**Hidden Parameter Block MDPs**

**Rating:** 6
**Confidence:** 2

**Review:**

This paper studies a family of Markov Decision Process (MDP) models with a low-dimensional unobserved state, called the block MDP. The authors assume that system dynamics of the underlying MDP is sufficiently summarized by a parameter $\theta$. This learning setting could be seen as a combination of Block MDP and the Hidden Parameter MDP; hence the name HiP-BMDP.

The authors then propose algorithms to learn parameters $\theta$ of HiP-BMDP from observed trajectories of the system. The authors derive the error bounds over between the Q-value parametrized by the learned $\theta$ and that of the actual parameter $\theta^*$. The authors then apply these results to the transfer learning settings where one applies the learned parameter $\theta$ to a different but somewhat similar environment parametrized by $\theta_i$. The error bound over Q-values in such a transfer learning settings is also provided. These results, including Theorems 2 and 3, seem sensible, while I haven't checked the details of the proof. Finally, the authors validate the proposed method through extensive simulations.

Overall, I believe this paper is well written and well organized. The assumptions of HiP-BMDP seem sensible and the derived error bounds look promising. These results are applicable in multi-taks learning settings. They explicate a set of parametric conditions under which one could accelerate the learning process of a future task from prior knowledge derived from previous tasks.

---

> ### Author Response · Authors · 2020-11-13
> **Anything to address?**
>
> Thank you for your review and kind words. Please do let us know if there is anything we can address for you to increase your score.

---

### Official Review · AnonReviewer2 · 2020-10-28
**Nicely connects two formalisms, but insight isn't clear and empirical results not convincing**

**Rating:** 7
**Confidence:** 4

**Review:**

Summary:
This paper combines representation learning via approximate bisimulation with the HiP-MDP framework for representing multiple tasks. The result is multi-task and meta-RL algorithms that operate from images and in the meta-RL case adapt to changes in the dynamics of the environment.

Pros:
Combines HiP-MDP and block MDP formalisms
Evaluates resulting method on a set of multi-task and meta-RL problems operating from image observations
Cons:
Not clear (to me) if any insight is gained from the theoretical analysis (the derivation of value and sample complexity bounds for approximate bisimulation was performed in Gelada et al. 2019)
Empirical gains are modest and it’s not clear if they are due to the image representation learning component of the loss or other aspects of the method.

Detailed Comments:
I think this paper would be relevant to cite: Learning an Embedding Space for Transferable Robot Skills (Hausman et al. 2018).

One thing that stood out to me was the very small number of training tasks used - only 4! I wonder if using more training tasks improves the generalization of single-task methods like DeepMDP that might be able to simply generalize over the dynamics changes given more examples. How much overfitting to the training tasks do you observe for these baseline methods versus HiP-BMDP?

It seems like there might be a baseline missing - generic multi-task algorithm that conditions on the environment ID but does use the bisimulation loss to help process the image observations. From my understanding, all baselines have no image representation learning component except DeepMDP which is not a multi-task or meta-RL algorithm.  I am a bit concerned that the gains we are seeing here have more to do with image representation learning than the structural assumptions employed by the algorithm.

I think that for the meta-RL setting, “environment ID” is replaced everywhere by experience seen so far in the task - is that correct? It would be good to make that clear, or if environment IDs are being used in meta-RL, how they are being used.

I’m a bit confused why there’s a gap between your method in the multi-task experiments but not in the meta-RL ones. Do the meta-RL experiments use image observations? If so, how is the PEARL baseline adapted to handle images? If not, why not?

Why does HiP-BMDP handle the sticky observation setting better than the baselines? Isn’t it impossible for it to persist state information over time steps?

Recommendation:
Borderline
I think the strong point of this paper would be the theoretical contribution of connecting bisimulation to HiP-MDPs. As I’m not very familiar with bisimulation theory, I cannot comment with confidence on the value of this contribution. It is not clear to me what new insights are gained from this extension.
Empirically, I am not convinced that the structural assumptions on the MDP used by the proposed algorithm yield performance improvement. For example, one assumption is that the reward function won’t change across tasks, but the method doesn’t really outperform a baseline that doesn’t make that assumption. I am concerned that the gains that are observed come from image representation learning via approximate bisimulation, and that this combined with any multi-task or meta-RL algorithm might achieve the same results.

---**Update**---
Increased score 5 -> 7 thanks to clarifications from authors.

---

> ### Author Response · Authors · 2020-11-16
> **Clarification of empirical results and insights**
>
> Thank you for your detailed comments and thoughtful review! To address your main concerns -- 1) insights: the bisimulation theory was used to design an objective for learning the hidden parameter in HiP-MDPs, which is our task bisimulation loss. This allows for our theoretical results -- the transfer and generalization bounds which extend beyond the results in the Gelada paper to handle different dynamics and the multi-task setting. The DeepMDP paper further does not have sample complexity bounds.  2) empirical results: We are not sure what baseline you are referring to that handles other reward functions but is not outperformed, as our method outperforms all other MTRL baselines in Fig 4. Could you clarify this concern further? Finally, the additional requested baseline is actually already given (specified in more detail below). We hope you will reconsider your score, given that we have clarified that the gains are not due to the image representation alone.
>
> Q: One thing that stood out to me was the very small number of training tasks used - only 4! I wonder if using more training tasks improves the generalization of single-task methods like DeepMDP that might be able to simply generalize over the dynamics changes given more examples. How much overfitting to the training tasks do you observe for these baseline methods versus HiP-BMDP?
> A: One of the goals of our work is to examine generalization capabilities in few environment settings, and how additional structural assumptions like HiP-MDP can improve few-shot generalization. Figure 9 in the appendix shows evaluation performance on the training environments, where we similarly see poorer performance by baselines, and therefore that overfitting is not yet happening for baselines.
>
>
> Q: It seems like there might be a baseline missing - generic multi-task algorithm that conditions on the environment ID but does use the bisimulation loss to help process the image observations. From my understanding, all baselines have no image representation learning component except DeepMDP which is not a multi-task or meta-RL algorithm. I am a bit concerned that the gains we are seeing here have more to do with image representation learning than the structural assumptions employed by the algorithm.
> A: We have clarified this in the paper, but the HiP-BMDP-nobisim baseline is this exact experiment! It is DeepMDP (what you are calling the bisimulation loss, aka a latent model loss and reward loss) without the “bisimulation loss” which we use to refer to our task embedding loss which using a form of bisimulation metric for the task embedding. Apologies for the confusion, we will rename our “bisimulation loss” to “task bisimulation metric loss” to better clarify the difference in objectives. This baseline shows that using the image representation learning component of DeepMDP with a task id is not sufficient to achieve the good performance exhibited by HiP-BMDP, and that our task bisimulation metric loss which leverages the structural assumptions of HiP-MDP is necessary.
>
>
> Q: I think that for the meta-RL setting, “environment ID” is replaced everywhere by experience seen so far in the task - is that correct? It would be good to make that clear, or if environment IDs are being used in meta-RL, how they are being used.
> A: Correct. The environment ID is not used in the meta-RL setting, and is replaced by the context constructed from the history. We use the same setup as PEARL.
>
>
> Q: I’m a bit confused why there’s a gap between your method in the multi-task experiments but not in the meta-RL ones. Do the meta-RL experiments use image observations? If so, how is the PEARL baseline adapted to handle images? If not, why not?
> A: The Meta-RL experiments do not use image observations, but proprioceptive state since all baseline algorithms are in that setting. Meta-RL techniques are too time intensive to train on pixel observations directly. We have updated the paper to make that more clear.
>
>
> Q: Why does HiP-BMDP handle the sticky observation setting better than the baselines? Isn’t it impossible for it to persist state information over time steps?
> A: Yes, what we are evaluating is the ability of each method to extract as much information from the current observation as possible and be robust to missing observations -- a relaxation of the Block MDP assumption. HiP-BMDP handles this setting better than other methods because is capturing all information available to predict future rewards in a principled manner through the bisimulation losses, unlike the other baselines.

---

> > ### Comment · AnonReviewer2 · 2020-11-23
> > **Thank you for the clarifications**
> >
> > Thank you for clarifying the “HiP-BMDP-nobisim” baseline. I think the renaming is a good idea, since the term “bisimulation” makes one think of the traditional definition of it, which does not involve the task.
> > The paper would be greatly strengthened if the meta-RL experiments were run with image observations, since the current results show no difference between your method and prior work. It appears that this CoRL 2020 paper was able to run existing meta-RL algorithms from images in simulation: https://arxiv.org/abs/2010.13957. It might be good to cite that paper as well since it also explores meta-RL from images.
> > I do also think it would be good to run experiments with more meta-training tasks. I suspect that with task distributions where the dynamics vary, a single non-adaptive agent trained across a set of training tasks is a very strong baseline when the number of tasks is large enough. You could still make the argument that your method works much better with fewer training tasks, which is more realistic.
> > Regarding the sticky observation question, thanks, I had forgotten about the reward prediction part of the bisimulation loss when I wrote that comment.
> >
> > I’m increasing my score to accept as I’m now confident that the results are not due to image representation learning alone. I encourage the authors to consider running the meta-RL experiments from images, as I think that will increase the impact of the paper.

---

> > > ### Author Response · Authors · 2020-11-24
> > > **Thanks for the response!**
> > >
> > > Thank you for reading our rebuttal and changing your score. And thank you for the reference to a meta-RL paper from pixels! We will try HiP-BMDP in this setting for a comparison to strengthen the paper.

---

### Official Review · AnonReviewer1 · 2020-11-03
**Good Paper**

**Rating:** 7
**Confidence:** 3

**Review:**

Hi,

First I want to thank authors for putting this together, I enjoyed reading it.

*Summary* Authors propose mixing Block MDP and Hidden Parameter MDP, and offer and algorithm (loss function) to learn this model that they claim to be useful for sample complexity and transferability among environments (that share state and action space). The proposed method showed promising results in experiments.

In general I enjoyed reading the paper, and I believe the idea is novel, a good step toward multi-task learning and well-written. My main concern is the validity of Block MDP assumption in real world.
1. Authors have to be more upfront on why this assumption is needed, it's hard now to exactly find why do we need Block MDP assumption, what would break if we were to relax this?
2. Authors have an experiment for this, but I would like to see how much p- probability of sticky observation would actually affect the performance, for example what would happen if we increase p, and what point the algorithm will break down? (Basically, I'm asking how sensitive the algorithm is to Block MDP assumption)

In general I enjoyed the paper, I'd like the theory part.
It seems like some notational clarity can help the paper, for example what is \Phi in the definition of \epsilon_T? Or what is Z in definition of \phi ?

---

> ### Author Response · Authors · 2020-11-20
> **Thank you for your assessment!**
>
> We thank the reviewer for their kind words and address concerns below and in the updated version of the paper.
>
> 1. Authors have to be more upfront on why this assumption is needed, it's hard now to exactly find why do we need Block MDP assumption, what would break if we were to relax this?
> A: A more extensive discussion of the Block MDP assumption has been added to the Background (Section 2). We also copy a similar version here for convenience:
>
> “The Block MDP assumption gives the Markov property in observation space, a key difference from partially observable MDPs (POMDPs), which have no guarantee of determining the generating state from the history of observations. This assumption allows us to compute reasonable bounds for our algorithm in k-order MDPs (which many real world problems can be described as, such as navigation, locomotion, and manipulation tasks from first person point of view) and avoids the intractability of true POMDPs, which have no guarantees on providing enough information to sufficiently predict future rewards. A relaxation of this assumption entails providing less information in the observation for predicting future reward, which will degrade performance.  We  show empirically that our method is still more robust to a relaxation of this assumption compared to other MTRL methods.”
>
> 2. Authors have an experiment for this, but I would like to see how much p- probability of sticky observation would actually affect the performance, for example what would happen if we increase p, and what point the algorithm will break down? (Basically, I'm asking how sensitive the algorithm is to Block MDP assumption).
> Weakening the Block MDP assumption breaks performance for all algorithms, because there is not enough information in the observation to predict information for any method. However, our experiments show that our method, HiP-BMDP, still outperforms other MTRL methods in this setting, which means it is better at extracting the limited information in the observation compared to other methods. We have added an additional plot to figure 7 showing the performance of HiP-BMDP under different values of p: 0.01, 0.02, 0.05, 0.1, 0.2, 0.5. Performance starts to decrease at p=0.1, and is completely unable to learn at p=0.5.
>
> 3. It seems like some notational clarity can help the paper, for example what is \Phi in the definition of \epsilon_T? Or what is Z in definition of \phi ?”
> A: Apologies, this definition was accidentally moved to appendix. It is now explained in the definition of \epsilon_T as: $\Phi T$ denotes the \textit{lifted} version of $T$, where we take the next-step transition distribution from observation space $\mathcal{X}$ and lift it to latent space $\mathcal{S}$. The $\mathbb{Z}$ in the definition of $\psi$ (I believe this is what you’re referring to, please correct me if I’m wrong) is the notation for the set of integers. We have clarified this to be the set of positive integers in the text.

---

### Decision · Program_Chairs · 2021-01-11
**Final Decision**

**Decision:**

Accept (Poster)

**Comment:**

The paper addresses the problem of learning and exploiting common (latent) task structure in multi-task reinforcement learning settings. The authors introduce a new formalism for capturing this type of structure and derive a gradient-based learning algorithm. They provide novel theoretical insights and strong empirical results.

Reviewers initially raised several concerns, regarding assumptions and especially accessibility of the paper (and in particular theoretical discussions). The majority of these concerns have been addressed in the detailed rebuttal. The resulting consensus is to accept the paper. Authors are encouraged to continue to improve accessibility of the paper for the camera ready submission.